# RELATIVE REPRESENTATIONS ENABLE ZERO-SHOT LATENT SPACE COMMUNICATION

**Luca Moschella**[1,*]   **Valentino Maiorca**[1,*]

**Marco Fumero**[1]   **Antonio Norelli**[1]   **Francesco Locatello**[2,†]   **Emanuele Rodolà**[1]

[1]Sapienza University of Rome   [2]Amazon Web Services

## ABSTRACT

Neural networks embed the geometric structure of a data manifold lying in a high-dimensional space into latent representations. Ideally, the distribution of the data points in the latent space should depend only on the task, the data, the loss, and other architecture-specific constraints. However, factors such as the random weights initialization, training hyperparameters, or other sources of randomness in the training phase may induce incoherent latent spaces that hinder any form of reuse. Nevertheless, we empirically observe that, under the same data and modeling choices, the angles between the encodings within distinct latent spaces do not change. In this work, we propose the latent similarity between each sample and a fixed set of anchors as an alternative data representation, demonstrating that it can enforce the desired invariances without any additional training. We show how neural architectures can leverage these relative representations to guarantee, in practice, invariance to latent isometries and rescalings, effectively enabling latent space communication: from zero-shot model stitching to latent space comparison between diverse settings. We extensively validate the generalization capability of our approach on different datasets, spanning various modalities (images, text, graphs), tasks (e.g., classification, reconstruction) and architectures (e.g., CNNs, GCNs, transformers).

## 1 INTRODUCTION

Neural Networks (NN) learn to transform high dimensional data into meaningful representations that are helpful for solving downstream tasks. Typically, these representations are seen as elements of a vector space, denoted as latent space, which corresponds to the constrained output (explicitly or implicitly) of a key component of the NN, e.g., the bottleneck in an Autoencoder (AE), or the word embedding space in an NLP task. The underlying assumption is that the learned latent spaces should be an optimal encoding given the data distribution, the downstream task, and the network constraints.

In practice, however, the learned latent spaces are subject to changes even when the above assumptions remain fixed. We illustrate this phenomenon in Figure 1, where we show the latent spaces produced by an AE with a two-dimensional bottleneck, trained on the `MNIST` dataset several times from scratch. These spaces differ from one another, breaking the fundamental assumptions made above. The distribution of the latent embeddings is affected by several factors, such as the random initialization of the network weights, the data shuffling, hyperparameters, and other stochastic processes in the training phase. Although different, the learned representations in Figure 1 are *intrinsically* similar: the distances between the embedded representations are approximately the same across all spaces, even if their absolute coordinates differ. Indeed, the learned latent spaces are the same up to a nearly isometric transformation.[1]

This symmetry is a consequence of the implicit biases underlying the optimization process (Soudry et al., 2018) forcing the model to generalize and, therefore, to give similar representations to similar samples with respect to the task. There exist infinitely many spatial arrangements complying with these similarity constraints, each associated with a different isometry in the example of Figure 1.

---

[*]Equal contribution.    [†] Work done outside of Amazon.
[1]To the best of our knowledge, the first to acknowledge this behavior was Olah (2015) in a blogpost.

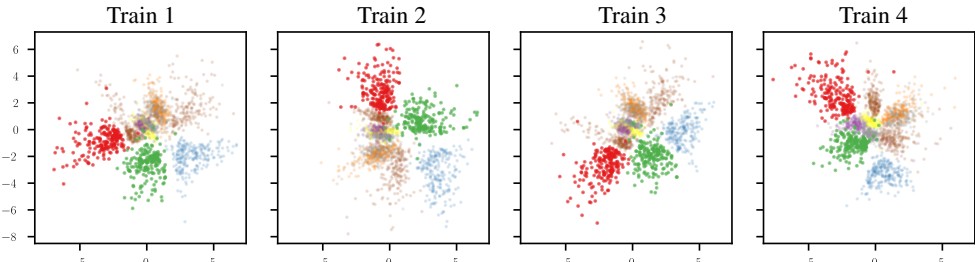

Figure 1: Latent spaces learned by distinct trainings of the same AE on the `MNIST` dataset. The bottleneck has size 2, thus there is no dimensionality reduction involved in the visualization of the latent space. The stochasticity in the training phase induces intrinsically similar representations. As we show in Figure 5, this property holds even for high-dimensional latent spaces.

But while the resulting models will be equally good in terms of the task, one still encounters several practical problems. For example, it is notoriously challenging to compare latent spaces across different trainings or across different NNs; perhaps more importantly, re-using neural components trained on different embeddings of the same data becomes impossible, since they are incompatible. To overcome this, we propose adopting a local coordinate system defined by the data itself. Each data point becomes a set of coefficients that encode the point as a function of other data samples, instead of an independent point in $\mathbb{R}^d$. The proposed *relative* representation directly encodes the intrinsic information underlying the data, and only depends on the angles between embeddings by construction. Remarkably, this enables a form of compositionality between learning models; it allows, for instance, to stitch together an encoder trained on ImageNet with a decoder trained on CIFAR, as we showcase in our experiments.

Our main contributions can be summarized as follows:

- We show that the representations learned by NNs are subject to change due to several training factors; nonetheless, the angles between latent embeddings are preserved.
- We introduce a novel relative representation for latent embeddings, that is invariant by construction to the transformations induced by stochastic factors in the training process.
- For the first time, we successfully demonstrate *zero-shot stitching* of neural components produced by distinct training regimens, e.g., due to different seeds or different neural architectures; we validate our findings on different data modalities (e.g. images, text).
- Our framework also provides a *quantitative* measure of performance while training neural models, which is differentiable, does not need any labeled data, and is correlated with standard performance measures such as accuracy.

## 2 RELATED WORK

**Representation similarity.** Recently, there has been growing agreement that good networks learn similar representations across a variety of architectures, tasks and domains (Morcos et al., 2018; Li et al., 2016; Kornblith et al., 2019; Bonheme & Grzes, 2022; Tsitsulin et al., 2020; Barannikov et al., 2022; Vulić et al., 2020; Lample et al., 2018; Lenc & Vedaldi, 2015; Mikolov et al., 2013b; Antonello et al., 2021), although this is still debated (Wang et al., 2018) and missing strong theoretical justifications. Similar observations have been made in the context of biological models Laakso & Cottrell (2000); Kriegeskorte et al. (2008); Raizada & Connolly (2012). Supported by the empirical evidence widely reported in these works, our method assumes that well-performing neural networks trained on similar tasks and data produce similar latent spaces, which allows us to define a representation that unifies all these spaces.

**Model stitching.** Lenc & Vedaldi (2015) introduced *trainable* stitching layers that allow swapping parts of different networks, while Bansal et al. (2021); Csiszárik et al. (2021) employed stitching to quantitatively verify statements such as "good networks learn similar representations" and "more data, width or time is better". Other works, such as Gygli et al. (2021); Biondi et al. (2021); Yaman et al. (2022); Bianchi et al. (2020), tried to directly produce compatible and reusable network

components without stitching layers; more generally, stitching has been adopted in the literature to analyze neural networks. In our work, we sidestep the need for trainable stitching layers and propose *zero-shot* model stitching to effectively reuse models.

**Relative information.** The attention mechanism (Vaswani et al., 2017) and its variants (Kossen et al., 2021) exploit the relationship between features to extract meaningful representations. Prototypical Networks (Snell et al., 2017) learn a metric space where the classification can be performed by measuring the distances to prototype representations. Shalam & Korman (2022) proposed the Self Optimal Transport feature transform to enrich the sample representations with higher order relations between the instance features, while Alvarez-Melis et al. (2019) proposed a general formulation of the optimal transport that accounts for global invariances in the underlying feature spaces. Mathematically, our method bears resemblance to a kernel method (Hofmann et al., 2008) as it employs inner products of embedded features as a core ingredient. However, differently from kernel methods, we do not introduce learnable parameters and, crucially, we compute the representations explicitly without resorting to a kernel trick.

## 3 METHOD

Given a training set $\mathbb{X}$, standard NNs learn an embedding function $E_\theta : \mathbb{X} \to \mathbb{R}^d$, parametrized by $\theta$, which maps each sample $\boldsymbol{x}^{(i)} \in \mathbb{X}$ to its latent representation, or **absolute representation**, $\boldsymbol{e}_{\boldsymbol{x}^{(i)}} = E_\theta(\boldsymbol{x}^{(i)})$. This representation is then exploited to solve downstream tasks, such as classification, reconstruction or generation, optimizing over some objective function of the general form:

$$\min_\theta \mathbb{E}_{x \in \mathbb{X}}[\mathcal{L}(E_\theta(x)) + Reg(\theta)]. \tag{1}$$

Here, $\mathbb{E}_{\mathbb{X}}$ denotes the expectation over the training distribution, and $Reg(\theta)$ encodes additional constraints on the weights $\theta$. As previously discussed, we argue that the learned weights $\theta^*$ are not only a function of $\mathbb{X}$ and of the specific loss appearing in Equation 1, but in practice they are also affected by the optimization process used to train the network due to weight initialization, data shuffling, hyperparameters, and other stochastic factors. We denote these factors collectively by $\phi$. In particular, as shown in Figure 1, changing these factors induces a transformation $T$ over the latent space, i.e., $\phi \to \phi'$ implies $E_\theta(\boldsymbol{x}^{(i)}) \to TE_\theta(\boldsymbol{x}^{(i)})$. We make the core assumption that $T$ preserves the angles between elements of the latent space, namely $\angle(\boldsymbol{e}_{\boldsymbol{x}^{(i)}}, \boldsymbol{e}_{\boldsymbol{x}^{(j)}}) = \angle(T\boldsymbol{e}_{\boldsymbol{x}^{(i)}}, T\boldsymbol{e}_{\boldsymbol{x}^{(j)}})$ for every $(\boldsymbol{x}^{(i)}, \boldsymbol{x}^{(j)}) \in \mathbb{X}$. While this assumption might seem too restrictive, in practice it arises in several real scenarios as we show in the following sections.

### 3.1 RELATIVE REPRESENTATIONS

To build our representation, we start by selecting a subset $\mathbb{A}$ of the training data $\mathbb{X}$, which we denote as **anchor** samples. Every sample in the training distribution will be represented with respect to the embedded anchors $\boldsymbol{e}_{\boldsymbol{a}^{(j)}} = E(\boldsymbol{a}^{(j)})$ with $\boldsymbol{a}^{(j)} \in \mathbb{A}$. As a measure capturing the relation between the anchors and the other samples, we consider a generic similarity function $sim : \mathbb{R}^d \times \mathbb{R}^d \to \mathbb{R}$, yielding a scalar score $r$ between two absolute representations $r = sim(\boldsymbol{e}_{\boldsymbol{x}^{(i)}}, \boldsymbol{e}_{\boldsymbol{x}^{(j)}})$. Given the anchors $\mathbb{A}$ in an arbitrary ordering $a^{(1)}, \ldots, a^{(|\mathbb{A}|)}$, we define the **relative representation** of $\boldsymbol{x}^{(i)} \in \mathbb{X}$ as:

$$\boldsymbol{r}_{\boldsymbol{x}^{(i)}} = (sim(\boldsymbol{e}_{\boldsymbol{x}^{(i)}}, \boldsymbol{e}_{\boldsymbol{a}^{(1)}}), sim(\boldsymbol{e}_{\boldsymbol{x}^{(i)}}, \boldsymbol{e}_{\boldsymbol{a}^{(2)}}), \ldots, sim(\boldsymbol{e}_{\boldsymbol{x}^{(i)}}, \boldsymbol{e}_{\boldsymbol{a}^{(|\mathbb{A}|)}})). \tag{2}$$

Figure 2 illustrates the key differences between absolute and relative representations.

**Choice of the anchors.** Anchors directly affect the expressivity of the relative representation space, and are related to the task at hand. For example, in a classification task, we should sample anchors from each class in the training set, in order to well represent each data sample in $\mathbb{X}$.

One case of interest arises when the data comes from different domains or modalities $\mathbb{X}$, $\mathbb{Y}$, and we are given a partial correspondence $\Gamma : P_\mathbb{X} \mapsto P_\mathbb{Y}$ mapping from a subset of $\mathbb{X}$ to a subset of $\mathbb{Y}$. In this case, we can sample anchors $\mathbb{A}_\mathbb{X} \subseteq P_\mathbb{X}$ and obtain corresponding anchors on the other domain directly as $\Gamma(\mathbb{A})$ (Norelli et al., 2022). We refer to these as *parallel anchors*. We show an example of parallel anchors in Section 5.2, where $\mathbb{X}$ and $\mathbb{Y}$ are Amazon reviews in two different languages.

The choice of the anchors is not restricted to elements in the training distribution. Given an encoder pre-trained on a fixed training distribution, we can pick elements from a set $\tilde{\mathbb{A}}$ that is out-of-domain

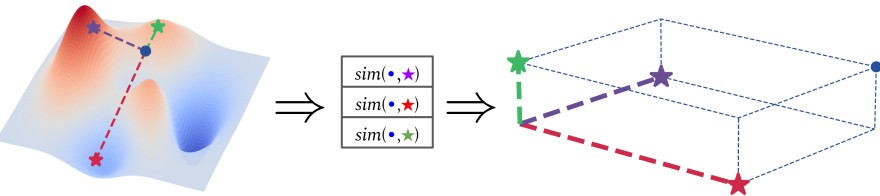

Figure 2: *Left*: Three anchors (colored stars) are selected on the data manifold; given a point on the manifold (blue dot), we compute its similarity w.r.t. the three anchors, yielding a vector of dimensionality 3 (middle). *Right*: Each dimension is treated as coefficients in a coordinate system defined by the anchors. Anchors are orthogonal in this example only for visualization purposes.

w.r.t. $\mathbb{X}$, and build the relative representations on top of $\tilde{\mathbb{A}}$. We refer to these as *OOD anchors* and exploit them, e.g., to solve domain adaptation tasks where we do not have access to a correspondence, and have scarce data labels. We refer again to the Sections 5.2 and 5.3 for real-world examples and to Appendix A.2 for a preliminary analysis of different selection strategies.

**Achieving latent invariance.** In this work, we choose the cosine similarity as the similarity function due to the properties it induces on the relative representation. The cosine similarity $S_C$ is the dot product of unit vectors, corresponding to the cosine of the angle $\theta$ between the two:

$$S_C(\boldsymbol{a}, \boldsymbol{b}) = \frac{\boldsymbol{a}\boldsymbol{b}}{||\boldsymbol{a}||||\boldsymbol{b}||} = \cos\theta\,. \tag{3}$$

Importantly, $\cos\theta$ does not change if we apply the same angle-preserving transformation $\boldsymbol{T}$ to two vectors $\boldsymbol{a}$ and $\boldsymbol{b}$, i.e., the cosine similarity is invariant to rotations, reflections, and rescaling. While this is not true for translations, NNs commonly employ normalization techniques (e.g., InstanceNorm (Ulyanov et al., 2016)) to center the latent spaces. Under this assumption, cosine similarity guarantees a relative representation $\boldsymbol{r}_{\boldsymbol{x}^{(i)}}$ invariant to angle-preserving transformations.

This means we have the freedom to change the embedding function $E_\theta$ with any other function $\tilde{E}$ that produces different representations with same angles, i.e.:

$$[S_C(\boldsymbol{e}_{\boldsymbol{x}^{(i)}}, \boldsymbol{e}_{\boldsymbol{a}^{(1)}}), \dots, S_C(\boldsymbol{e}_{\boldsymbol{x}^{(i)}}, \boldsymbol{e}_{\boldsymbol{a}^{(|\mathbb{A}|)}})] = [S_C(\tilde{\boldsymbol{e}}_{\boldsymbol{x}^{(i)}}, \tilde{\boldsymbol{e}}_{\boldsymbol{a}^{(1)}}), \dots, S_C(\tilde{\boldsymbol{e}}_{\boldsymbol{x}^{(i)}}, \tilde{\boldsymbol{e}}_{\boldsymbol{a}^{(|\mathbb{A}|)}})]\,, \tag{4}$$

where $\tilde{\boldsymbol{e}}_{\boldsymbol{x}^{(i)}} = \tilde{E}(\boldsymbol{x}^{(i)}) = \boldsymbol{T}E(\boldsymbol{x}^{(i)})$ and $\boldsymbol{T}$ is an arbitrary angle-preserving transformation. A practical application of this invariance is the possibility of comparing latent spaces across multiple trainings, and re-using models as demonstrated in Sections 4 and 5.

We remark that other choices of similarity function can be made to enforce different invariances into the representation. For example, one may impose invariance to non-isometric deformations with bounded distortion. We did not find this to be necessary in our experiments, as typically NNs that generalize sufficiently well can handle small perturbations. Nevertheless, this invariance can be enforced by design with vector quantization algorithms. Figure 7 shows a preliminary investigation, leaving further exploration to future work.

## 4 LATENT SPACE COMMUNICATION

In this section, we demonstrate how our relative representations can effectively be used to produce latent spaces that are stable under a variety of factors. Our main question is the following: Given two different learning models that are trained separately on different data, can we compare their latent embeddings? In asking this, we assume that the two models are trained on a similar phenomenon, e.g., on two different samplings of the English language or on two different modalities.

We answer in the positive, showing the gained invariance enables effective communication between different, but semantically equivalent latent spaces. In particular, we analyze how different word embedding spaces, once projected onto relative representations, are intrinsically the same (Section 4.1); we then show how the similarity between the relative counterparts of two or more embedding spaces is a surprisingly good predictor of model performance (Section 4.2); finally, we confirm that relative representations in the training phase are not detrimental to performance (Section 4.3).

## 4.1 WORD EMBEDDINGS

**Experimental setting.** We select two different word embeddings on the English language, namely `FastText` (Bojanowski et al., 2017) and `Word2Vec` (Mikolov et al., 2013a). Both models are pre-trained on different data, but partly share a vocabulary from which we extract $\approx 20\text{K}$ words. Using 300 randomly drawn parallel anchor, we convert each embedding space to a relative one. In Table 1 (left), we show the original and the relative embeddings. For each word $w$, we consider its corresponding encodings $x$ and $y$ in the source and target space. We apply three different metrics to measure their similarity (in a setting similar to Vulić et al. (2020)): (i) *Jaccard*: the discrete Jaccard similarity between the set of word neighbors of $x$ in source and target; (ii) *Mean Reciprocal Rank*: measures the (reciprocal) ranking of $w$ among the top-k neighbors of $x$ in the target space; (iii) *Cosine*: measures the cosine similarity between $x$ and $y$. Additional details in Appendix A.5.1

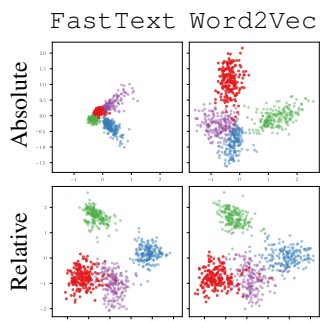

| | Source | Target | Jaccard ↑ | MRR ↑ | Cosine ↑ |
|---|---|---|---|---|---|
| **Absolute** | FT | FT | $1.00 \pm 0.00$ | $1.00 \pm 0.00$ | $1.00 \pm 0.00$ |
| | | W2V | $0.00 \pm 0.00$ | $0.00 \pm 0.00$ | $0.01 \pm 0.00$ |
| | W2V | FT | $0.00 \pm 0.00$ | $0.00 \pm 0.00$ | $0.01 \pm 0.00$ |
| | | W2V | $1.00 \pm 0.00$ | $1.00 \pm 0.00$ | $1.00 \pm 0.00$ |
| **Relative** | FT | FT | $1.00 \pm 0.00$ | $1.00 \pm 0.00$ | $1.00 \pm 0.00$ |
| | | W2V | $0.34 \pm 0.01$ | $0.94 \pm 0.00$ | $0.86 \pm 0.00$ |
| | W2V | FT | $0.39 \pm 0.00$ | $0.98 \pm 0.00$ | $0.86 \pm 0.00$ |
| | | W2V | $1.00 \pm 0.00$ | $1.00 \pm 0.00$ | $1.00 \pm 0.00$ |

Table 1: Qualitative *(left)* and quantitative *(right)* comparisons of English word embeddings using absolute and relative representations. PCA is applied only for visualization. All metrics are calculated with $K = 10$ averaged over 20k words and across 10 different random seeds. See Figure 10 for other dimensionality reductions and Table 8 and fig. 11 for the same experiment on `CIFAR-10`.

**Result analysis.** Table 1 *(left)* highlights clusters of semantically similar words and shows that the absolute representations are incoherent across the two latent spaces, while the relative embeddings are highly similar. The average Jaccard distance reported in Table 1 *(right)*, says that the word neighborhoods of the relative representations are matched exactly 34% of the time in one direction, and 39% of the time in the other one (the missing 61% is due to semantic differences, that are not taken into account by the discrete nature of the Jaccard metric). By contrast, the absolute embeddings are never matched exactly (Jaccard score equal to zero); for a match to happen, it would mean that the `FastText` and `Word2Vec` embeddings of a given English word are almost the same, which is highly unlikely. MRR, close to a perfect score for the relative representations, shows that the most-similar word to a given one is usually itself, even if their cosine similarity doesn't reach 1.

Overall, these results show that relative representations are preserved across different word embedding models, validating our assumptions.

## 4.2 LATENT DISTANCE AS A PERFORMANCE PROXY

**Experimental setting.** In this experiment, we consider a node classification task on the `Cora` graph dataset (Sen et al., 2008). We first train a *reference* model that achieves good accuracy on a validation set. Then, we train $\approx 2000$ models with various combinations of seed, number of epochs, number of layers, dropout probability, activation functions, optimizer type, learning rate or type of graph embedder (Table 10). All the models are classically trained using absolute representations, which are converted to relative post-training by projecting the embeddings onto 300 randomly drawn but fixed anchors. For each model, we measure its classification accuracy and compute the similarity of its space with the reference one. This similarity is computed as the average cosine similarity between the node embeddings produced by a given model and the corresponding embeddings in the reference.

**Result analysis.** The scatter plot in Figure 3 *(left)* shows that better-performing models tend to be the ones with the latent spaces most similar to the reference. The performance-similarity correlation

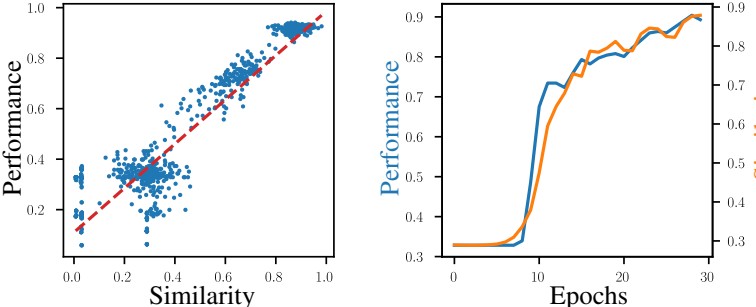

Figure 3: Graph node classification task on `Cora`. *Left:* Correlation between the performance of $\approx 2000$ models and the similarity of their latent spaces with respect to a well-performing reference model. *Right:* The same correlation plotted over time. The mean Pearson correlation over all models is $0.955$, after filtering out the models having best validation accuracy below $0.5$.

also holds over time, as shown in Figure 3 *(right)*. Additional correlation examples are in Figure 9. Interestingly, this metric is differentiable, enabling an explicit supervision signal on the latent space, which does not require labeled data and could be readily exploited in a teacher-student framework.

Overall, these results suggest that the similarity between the relative representations of latent spaces is a remarkably good proxy to evaluate model performance.

### 4.3 TRAINING WITH ABSOLUTE VS. RELATIVE REPRESENTATIONS

**Experimental setting.** Finally, we compare architectures that do or do not employ the relative representation while training. In these experiments, the models vary slightly according to the dataset; however, the relative and absolute versions are always comparable in terms of architecture, number of learnable parameters and hyperparameters. We refer to the supplementary material and the open-source code for further details on their implementation. In this section we consider classification tasks on several datasets, spanning the image domain (Lecun et al., 1998; Xiao et al., 2017; Krizhevsky, 2009) and the graph domain (Yang et al., 2016).

Table 2: Performance comparison between relative and absolute representations on several image and graph datasets. The metric is the classification weighted F1 score ($\pm$ std), over 6 seeds.

| | Image Classification | | | | Graph Node Classification | | |
|---|---|---|---|---|---|---|---|
| | MNIST | F-MNIST | CIFAR-10 | CIFAR-100 | Cora | CiteSeer | PubMed |
| **Relative** | $97.91 \pm 0.07$ | $90.19 \pm 0.27$ | $87.70 \pm 0.09$ | $66.72 \pm 0.35$ | $0.89 \pm 0.02$ | $0.77 \pm 0.03$ | $0.91 \pm 0.01$ |
| **Absolute** | $97.95 \pm 0.10$ | $90.32 \pm 0.21$ | $87.85 \pm 0.06$ | $68.88 \pm 0.14$ | $0.90 \pm 0.01$ | $0.78 \pm 0.03$ | $0.91 \pm 0.01$ |

**Result analysis.** The results, reported in Table 2, show that relative representations, when used at training time, are not detrimental to performance in general. This is further shown in Tables 3 to 6 and 15 to 18, where a subset of the results compares the absolute and relative representations on a variety of domains, datasets and tasks.

Overall, these results show that relative representations are effective when involved in end-to-end training, without significant performance drops.

## 5 ZERO-SHOT MODEL STITCHING

In this section, we illustrate how the latent space communication demonstrated in Section 4 enables zero-shot interoperability of pre-trained neural components. In previous works, such as Lenc & Vedaldi (2015); Bansal et al. (2021), stitching layers are *trainable* linear projections that allow swapping parts of different networks. Instead, on relative representations unlocks the possibility of *zero-shot stitching* different neural components, treating them as frozen black-box modules.

We define a generic *stitched model* as the composition of an encoder, that embeds data, plus a *relative* decoder specialized in a downstream task (classification, reconstruction). The stitching operation is always performed without training or fine-tuning, in a zero-shot fashion. Hereafter, we showcase stitching capabilities across combinations of different stochasticity sources (Figure 4 and table 3), neural architectures (Tables 4 and 5) or datasets (Table 6). Finally, we present strong real-world applications in NLP (Section 5.2) and CV (Section 5.3), e.g. zero-shot predictions on novel languages. Additional implementation details are given in the supplementary materials.

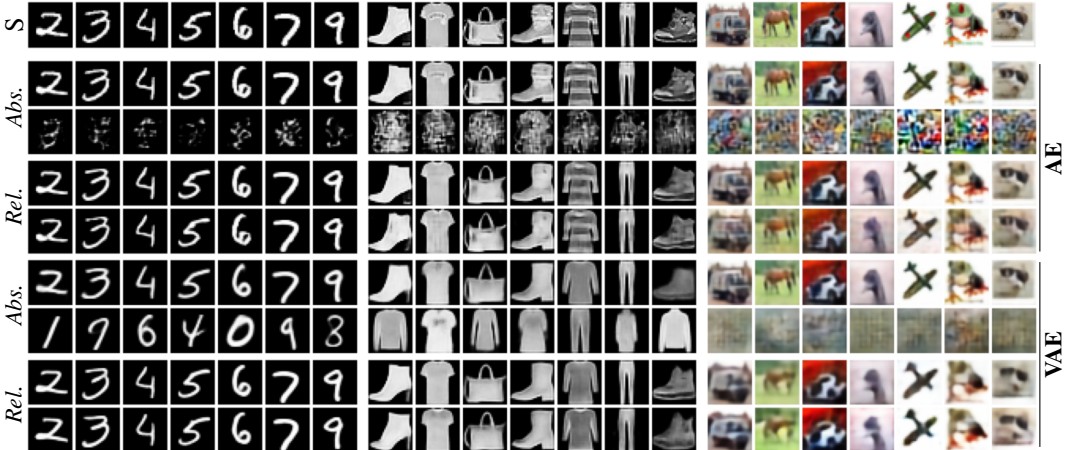

Figure 4: Reconstruction examples. Each column is a different image, row pairs are different architectures. In each pair, we first report the non-stitched reconstructions, then the stitched ones.

## 5.1 IMAGE RECONSTRUCTION

**Experimental setting.** We perform zero-shot stitching with AEs and VAEs trained with relative representations end-to-end on several datasets. For each combination of model and dataset, we perform 5 trainings with different seeds, and zero-shot stitch together the resulting encoders and decoders.

**Result analysis.** In Figure 4 the stitched models that employ absolute representations (*Abs.*) produce erroneous predictions, since the latent spaces obtained from distinct trainings are incompatible. Interestingly, although the absolute VAE does not produce compatible latent spaces, it is regularized, thus all embeddings produced by the encoders correspond to wrong but semantically meaningful reconstructions. Relative representations (*Rel.*) exhibit almost indistinguishable reconstructions between the models trained end-to-end and the stitched ones. Quantitative results are in Table 3.

These results support our claim that relative representations are empirically invariant to training stochasticity.

Table 3: Stitching performance. The MSE ($\pm$ std) between the ground truth $\mathbb{X}$ and the reconstructions is computed over 5 different seeds. Stitching with our relative representations yields an error up to two orders of magnitude less than the absolute counterpart.

| | | | MNIST | F-MNIST | CIFAR-10 | CIFAR-100 | MSE $\downarrow$ |
|---|---|---|---|---|---|---|---|
| AE | Abs. | Non-Stitch. | $0.66 \pm 0.02$ | $1.57 \pm 0.03$ | $1.94 \pm 0.08$ | $2.13 \pm 0.08$ | $1.58 \pm 0.05$ |
| | | Stitch. | $97.79 \pm 2.48$ | $120.54 \pm 6.81$ | $86.74 \pm 4.37$ | $97.17 \pm 3.50$ | $100.56 \pm 4.29$ |
| | Rel. | Non-Stitch. | $1.18 \pm 0.02$ | $3.59 \pm 0.04$ | $2.83 \pm 0.13$ | $3.50 \pm 0.08$ | $2.78 \pm 0.07$ |
| | | Stitch. | $2.83 \pm 0.20$ | $6.37 \pm 0.29$ | $5.39 \pm 1.18$ | $18.03 \pm 12.46$ | $8.16 \pm 3.53$ |
| VAE | Abs. | Non-Stitch. | $1.31 \pm 0.04$ | $4.38 \pm 0.03$ | $2.68 \pm 0.06$ | $3.00 \pm 0.03$ | $2.84 \pm 0.04$ |
| | | Stitch. | $98.51 \pm 1.49$ | $118.96 \pm 2.96$ | $69.02 \pm 1.54$ | $78.57 \pm 1.88$ | $91.27 \pm 1.97$ |
| | Rel. | Non-Stitch. | $2.97 \pm 0.14$ | $6.81 \pm 0.06$ | $5.18 \pm 0.22$ | $5.93 \pm 0.14$ | $5.22 \pm 0.14$ |
| | | Stitch. | $13.43 \pm 6.79$ | $24.03 \pm 13.15$ | $11.20 \pm 3.15$ | $11.23 \pm 2.38$ | $14.97 \pm 6.37$ |

## 5.2 TEXT CLASSIFICATION

In this Section, we show practical examples of the use of parallel anchors (Sec 3.1).

Table 4: Cross-lingual stitching performance comparison. The table reports the mean weighted F1 ($\pm$ std) and MAE on `Amazon Reviews` coarse-grained, across 5 seeds.

| | | Absolute | | Relative | | | |
| | | | | Translated | | Wikipedia | |
| Decoder | Encoder | FScore | MAE | FScore | MAE | FScore | MAE |
|---|---|---|---|---|---|---|---|
| en | en | $91.54 \pm 0.58$ | $0.08 \pm 0.01$ | $90.06 \pm 0.60$ | $0.10 \pm 0.01$ | $90.45 \pm 0.52$ | $0.10 \pm 0.01$ |
| | es | $43.67 \pm 1.09$ | $0.56 \pm 0.01$ | $82.78 \pm 0.81$ | $0.17 \pm 0.01$ | $78.53 \pm 0.30$ | $0.21 \pm 0.00$ |
| | fr | $54.41 \pm 1.61$ | $0.45 \pm 0.02$ | $78.49 \pm 0.66$ | $0.21 \pm 0.01$ | $70.41 \pm 0.57$ | $0.29 \pm 0.01$ |
| | ja | $48.72 \pm 0.90$ | $0.51 \pm 0.01$ | $65.72 \pm 0.55$ | $0.34 \pm 0.01$ | $66.31 \pm 0.80$ | $0.34 \pm 0.01$ |

Table 5: Cross-architecture stitching performance comparison. The table reports the mean weighted F1 ($\pm$ std) for each dataset, across 5 different seeds.

| | | TREC | DBpedia | Amazon Reviews | |
| | | | | Coarse | Fine |
|---|---|---|---|---|---|
| Abs. | Non-Stitch | $91.70 \pm 1.39$ | $98.62 \pm 0.58$ | $87.81 \pm 1.58$ | $55.35 \pm 3.19$ |
| | Stitch | $21.49 \pm 3.64$ | $6.96 \pm 1.46$ | $49.58 \pm 2.95$ | $19.01 \pm 2.04$ |
| Rel. | Non-Stitch | $88.08 \pm 1.37$ | $97.42 \pm 2.05$ | $85.08 \pm 1.93$ | $48.92 \pm 3.57$ |
| | Stitch | $75.89 \pm 5.38$ | $80.47 \pm 21.14$ | $72.37 \pm 7.32$ | $33.24 \pm 7.21$ |

**Experimental setting.** We consider two different text classification settings.

*Cross-lingual*: given a review predict the associated star rating, done on multi-lingual data from the `Amazon Reviews` dataset (Keung et al., 2020). Following the original paper, we work on a binarized version of the task, with FScore and MAE as metrics. In the supplementary material, we report results on the fine-grained formulation. We adopt four different pre-trained language-specific RoBERTa transformers (Liu et al., 2019) and evaluate their zero-shot stitching performance on languages never seen by the classifier. We use parallel anchors in two modalities: i) *Translated*: consider English reviews translated[2] into the other languages; ii) *Wikipedia*: adopt an external corpus, WikiMatrix (Schwenk et al., 2021), providing parallel sentences extracted from Wikipedia.

*Cross-architecture*: assessed on three different datasets: `TREC` (coarse) (Hovy et al., 2001), `DBpedia` (Zhang et al., 2015), `Amazon Reviews` (English split). We adopt two different pre-trained BERT (Devlin et al., 2019) transformers (cased and uncased version), ELECTRA (Clark et al., 2020) and RoBERTa.

**Result analysis.** Tables 4 and 5 show for the first time that it is possible to learn to solve a downstream task on a specific language or transformer and perform predictions on another.

Stitching with absolute representations yields performances comparable to random guessing across the board, proving that relative representations are a key element for the success of this kind of zero-shot stitching. Moreover, Table 4 highlights the robustness that relative representations have on the choice of anchors, even when they are noisy (*Translated* case), or their distribution differs from the one of the downstream task (*Wikipedia* case), as long as their encoding can be handled correctly by the encoder. In our case, the encoder is pre-trained to represent a variety of texts in a specific language, thus, even if WikiMatrix has a completely different domain from `Amazon Reviews`, the transformer still computes a meaningful and comparable representation with those of the reviews. We report in Tables 15 and 16 complete results on all languages combination, and in Table 17 the performance obtained by a multi-lingual transformer. To the best of our knowledge, it is the only alternative for obtainining compatible representations across languages.

According to these results, relative representations show invariance to different architectures and data distribution shifts (e.g., different train languages).

---

[2]We used the `=GOOGLETRANSLATE` function available in Google Sheets.

## 5.3 IMAGE CLASSIFICATION

In this Section, we show practical examples of the use of OOD anchors (Sec 3.1).

Table 6: Stitching performance comparison with different encoding techniques. The table reports the mean weighted F1 ($\pm$ std) on `CIFAR-100` coarse-grained and `ImageNet1k`, across 5 seeds.

| Decoder | Encoder | CIFAR-100 | | ImageNet1k | |
|---|---|---|---|---|---|
| | | Absolute | Relative | Absolute | Relative |
| rexnet-100 | rexnet-100 | $82.06 \pm 0.15$ | $80.22 \pm 0.28$ | $73.78 \pm 0.29$ | $72.61 \pm 0.16$ |
| | vit-base-patch16-224 | - | $54.98 \pm 0.44$ | - | $37.39 \pm 0.36$ |
| | vit-base-resnet50-384 | - | $53.33 \pm 0.37$ | - | $42.36 \pm 0.36$ |
| | vit-small-patch16-224 | - | $59.82 \pm 0.32$ | - | $43.75 \pm 0.27$ |
| vit-base-patch16-224 | rexnet-100 | - | $76.81 \pm 0.49$ | - | $30.78 \pm 0.81$ |
| | vit-base-patch16-224 | $93.15 \pm 0.05$ | $91.94 \pm 0.10$ | $80.91 \pm 0.29$ | $78.86 \pm 0.33$ |
| | vit-base-resnet50-384 | $6.21 \pm 0.33$ | $81.42 \pm 0.38$ | $0.07 \pm 0.05$ | $44.72 \pm 0.57$ |
| | vit-small-patch16-224 | - | $84.29 \pm 0.86$ | - | $48.31 \pm 0.72$ |
| vit-base-resnet50-384 | rexnet-100 | - | $79.79 \pm 0.43$ | - | $53.46 \pm 0.68$ |
| | vit-base-patch16-224 | $4.69 \pm 0.07$ | $84.46 \pm 0.19$ | $0.08 \pm 0.04$ | $62.21 \pm 0.54$ |
| | vit-base-resnet50-384 | $91.41 \pm 0.09$ | $90.77 \pm 0.16$ | $82.55 \pm 0.30$ | $81.88 \pm 0.16$ |
| | vit-small-patch16-224 | - | $84.66 \pm 0.16$ | - | $61.32 \pm 0.36$ |
| vit-small-patch16-224 | rexnet-100 | - | $75.35 \pm 0.41$ | - | $37.58 \pm 0.44$ |
| | vit-base-patch16-224 | - | $81.23 \pm 0.31$ | - | $50.08 \pm 0.63$ |
| | vit-base-resnet50-384 | - | $78.35 \pm 0.69$ | - | $45.45 \pm 1.41$ |
| | vit-small-patch16-224 | $90.07 \pm 0.19$ | $88.85 \pm 0.44$ | $77.73 \pm 0.41$ | $76.36 \pm 0.40$ |

**Experimental setting.** We consider a classification task on `ImageNet1k` and `CIFAR-100` with coarse labels (20), and 4 different pre-trained image encoders: three variants of the ViT transformer (Dosovitskiy et al., 2021) and RexNet (Han et al., 2020).

**Result analysis.** The results in Table 6 highlight how the relative representations allow stitching modules with different encoding dimensionality, since the decoder receives a relative representation with guaranteed equal size. Further, the results demonstrate the ability to generalize and perform zero-shot stitching on `CIFAR-100`, although that data was never seen by the encoder since it is a frozen transformer trained on `ImageNet1k`. Interestingly, `rexnet-100` is the only transformer whose latent dimensionality is higher than the number of anchors, and the biggest drop in stitching performance happens when the decoder is trained on it. This suggests the number of anchors is an important hyperparameter; we refer to Figure 6 for a deeper analysis.

Overall, these results prove that relative representations can bridge general-purpose encoders and pre-trained task-specific decoders.

## 6 CONCLUSION

In this work, we introduced the concept of relative representations to enable zero-shot latent space communication, with several practical consequences as showcased in our discussion and experiments. Our work proves that a latent semantic correspondence between data domains, when present, can be exploited through a simple representation shift, without resorting to sophisticated processing or heavy training.

**Limitations and future work.** Our work is open to several follow-up directions. While in this paper we considered the cosine similarity, different functions can enforce additional invariances in the relative representation. The study of invariant latent spaces as a general direction has the potential to lead to further impact; in Figure 7 we showed preliminary results of this possibility. Another interesting line of research to improve the representation expressivity would be to estimate *geodesic* distances over the data manifold instead of adopting Euclidean approximations. Similarly, we believe that the connections between the composition of the anchors set $\mathbb{A}$ and the expressivity of relative representations demands additional research. For example, the training cost is directly affected by the number and update frequency of the anchors. Finally, the stitching procedure may be extended to multiple layers, promoting reusable network components.

ACKNOWLEDGMENTS

The authors gratefully acknowledge the anonymous reviewers for the thoughtful remarks, and Luigi Gresele for the insightful discussions. This work is supported by the ERC Starting Grant No. 802554 (SPECGEO).

REPRODUCIBILITY STATEMENT

We describe in detail the relative representation computation in Section 3.1. We describe the experimental settings for the various scenarios, and refer to the supplementary material for further implementation details (Appendix A.5). Moreover, we release a well-documented and modular codebase, with the relative representation layer being implemented as a stand-alone PyTorch module. All the checkpoints used in the experiments are versioned with DVC (Kuprieiev et al., 2023) to easily reproduce all the figures and tables. The stand-alone module allows the integration of the relative representations in any existing neural network effortlessly.

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

# A  APPENDIX

## A.1  HIGH-DIMENSIONAL LATENT SPACES

In Figure 1, multiple trainings of the same two-dimensional AE produce intrinsically similar latent spaces; in Figure 5 we show this property also holds on AEs with a high-dimensional bottleneck. In the first row, PCA is fitted indipendently in each column, and since the PCA transformation produces the same output everywhere the latent spaces are intrinsically the same. In the second row, PCA is fitted only on the first latent space; since in this case the PCA transformation produces different outputs, the latent spaces, although intrinsically similar, are extrinsically different.

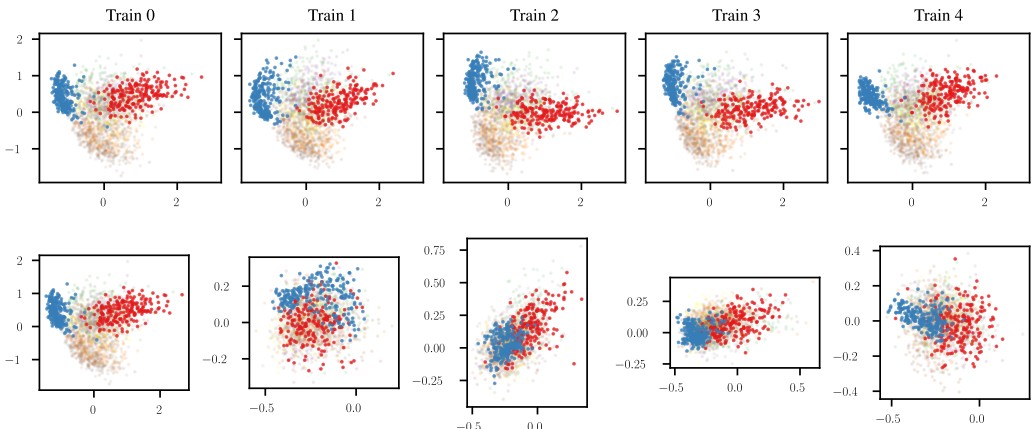

Figure 5: Latent spaces learned by distinct trainings of the same high-dimensional AE on the `MNIST` dataset. Each column is the latent space obtained by the AE with a different seed. On the first row, the dimensionality reduction is performed through PCAs fitted independently on each latent space, meanwhile, on the second row PCA is fitted on the leftmost latent space and then applied to all of them.

## A.2  ANCHORS ANALYSIS

The cardinality of the anchors set $\mathbb{A}$ and the choice of specific anchors is crucial to the quality of the relative representations. At the extreme, selecting one single anchor or the same repeated data points for all anchors, will produce collapsed relative representations. We believe that additional research is required to obtain a better understanding on the optimal choice for $\mathbb{A}$. Questions like "Are anchors set composed only by stopwords worse than the ones composed by meaningful and diverse words?" require empirical evidence and could help revealing the semantics of the latent space. Indeed, each anchor is associated with a dimension in a relative representation; one could inspect the anchor data point to get a sense of the meaning of that latent dimension.

**Anchor number.**    Below, we report a preliminary study on the performance sensitivity against the cardinality of the anchors set. In Figure 6 we report the performance on the node classification task on `Cora`, with a model trained end-to-end adopting the relative representations while training, and on image classification tasks on `CIFAR-100`, with a frozen encoder. The performance improves monotonically as the number of anchors increase when the absolute representations are frozen *(right)*. Differently, training models end-to-end proves to be more susceptible to model collapse and instabilities, as increasing the number of anchors does not always improve the performance *(left)*. Further research on the relation between the absolute latent space dimensionality and the relative representation dimensionality (i.e., the number of anchors) is needed to clarify how the two quantities impact the performance, when training end-to-end or not.

**Anchor selection.**    In Tables 7 and 8, we analyze different anchor selection strategies under an experimental setting analogous to the one described in Section 4.1:

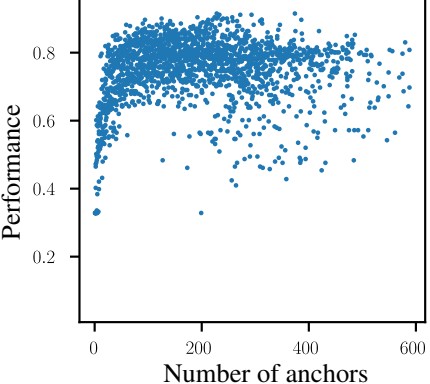 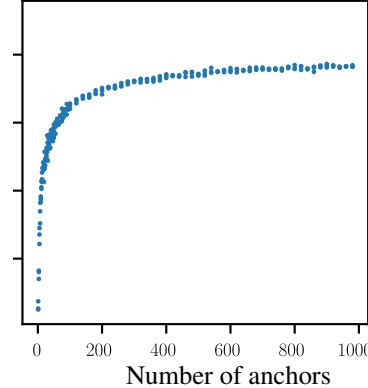

Figure 6: Accuracy vs Number of anchors. Each point is a trained model. *Left*: Trained embedder on Cora, node classification. *Right*: Frozen transformer on Cifar100 coarse-grained, image classification. Left is less stable because the absolute embeddings are trained, and we are working on a domain that is less stable (graphs). Some collapsed examples are not visualized.

- **uniform** The first selection strategy is the same adopted in the main manuscript. We randomly select the anchors with a uniform probability distribution over all the available samples;

- **fps** We select the anchors according to a farthest point sampling strategy;

- **kmeans** We select the anchors as the words more close to the centroids of K-means clustering with $K =$ number of anchors;

- **top$\{k\}$** We select the anchors as the $k$ most frequent words, after skipping the first 400 which are mostly stopwords.

We expect strategies that better cover the absolute space with anchors to be the most effective ones. Indeed, the results are comparable across selection strategies, but fps reaches everywhere the best Jaccard and MRR scores while k-means the best Cosine ones. We attribute this behavior to their different nature: they both rely on the geometry of the latent spaces they are applied to, but k-means also favors high-density regions, and this can become a negative bias for the task at hand. In general, the uniform sampling is the most straightforward to apply, since it does not require additional computation for the selection process, and still achieves good performances.

## A.3 INVARIANCE WITH GUARANTEED BOUNDS

In this section, we explore a slightly modified version of the similarity function adopted in the main paper. The experimental setting is the same as in Section 4.1. We want to measure the similarity between pairs of absolute embeddings and their relative counterparts. To get some kind of quantitative measure, we add a similarity score calculated as the pairwise cosine distance between the two embedding types, averaged. Therefore, a lower score indicates the spaces are more similar. On top of the standard relative representations, the ones computed with $sim = S_C$, here we try to improve the similarity measure with guaranteed robustness to bounded distortion. In Figure 7 we report preliminary results that adopt this technique: a vector-quantized similarity function produces relative representations which are more similar (they have a lower score). The vector-quantization is done through agglomerative clustering on the absolute embeddings at various thresholds $t$. We leave to future works the study of the trade-off between guaranteed invariance to arbitrary bounded distorsion and the expressiveness of the resulting representations.

## A.4 DATASET INFORMATION

In Table 9 we summarize the datasets utilized in our work, and for each one, we specify the number of classes, to give an idea about the classification difficulty.

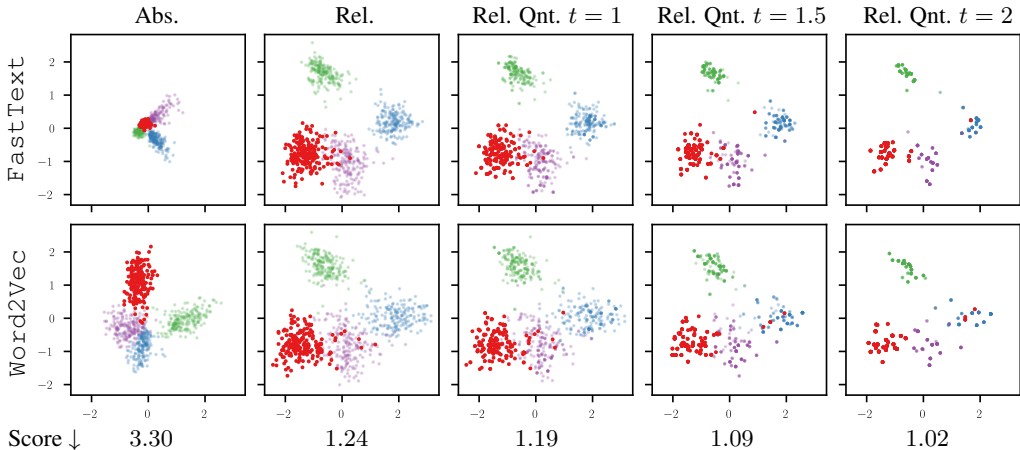

Figure 7: The `FastText` and `Word2Vec` embeddings of a subset of the English dictionary. The score is the pairwise distance average between the two embedding types, thus a lower score indicates the spaces are more similar. The absolute representations appear very dissimilar meanwhile the relative representations yield almost identical spaces. Quantizing the absolute representations by performing agglomerative clustering with distance threshold $t$ produces even more similar spaces.

## A.5 IMPLEMENTATION DETAILS

In this Section, following the corresponding sections in the main paper, we report implementation details for all the experimental settings considered.

**Tools & Technologies**  In all the experiments presented in this work, the following tools were used:

- *NN-Template* GrokAI (2021), to easily bootstrap the project and enforce best practices;
- *PyTorch Lightning* (Falcon & The PyTorch Lightning team, 2019), to ensure reproducible results while also getting a clean and modular codebase;
- *Weights and Biases* (Biewald, 2020), to log experiments and compare runs across huge sweeps;
- *Transformers by HuggingFace* (Wolf et al., 2020), to get ready-to-use transformers for both text and images;
- *Datasets by HuggingFace* (Lhoest et al., 2021), to access most of the NLP datasets and ImageNet for CV;
- *DVC* (Kuprieiev et al., 2023), for data versioning;
- *PyTorch Geometric* (Fey & Lenssen, 2019), to handle graph datasets and get ready-to-use GNN architectures.

### A.5.1 WORD EMBEDDINGS

For both the Figure and the Table in Section 4.1, the number of anchors is set to 300 for a fair comparison with the dimensionality of the original spaces. For visualization purposes, we needed the figure to both show an easy clusterable and restricted set of word embeddings. They are obtained by subsampling the shared vocabulary with the following procedure: we select 4 random pivot words, and for each of them we consider the top-200 words in their neighborhood. This results in a total of 800 points divided in 4 clusters, the ones used only for the visualization part. For the quantitative part (table results), we select 20K random words from the shared vocabulary with a fixed seed for reproducibility purposes.

For the computer vision counterpart (Figure 11 and table 8), the procedure is similar but with the following differences: i) the number of anchors is set to 500 to balance between the different encoding dimensions of the two transformers (384 for ViT-small and 768 for ViT-base); ii) the subsampling for visualization purposes is done by selecting 4 classes and randomly picking 200 samples for each of them;

**Evaluation metrics** Consider the set of $\approx$ 20k samples $\mathbb{S}$ (words for the NLP test, images for the CV one) and the source space $\mathbb{X}$ and target space $\mathbb{Y}$ and any sample $s \in \mathbb{S}$, we compute its representation in $\mathbb{X}$ and $\mathbb{Y}$ through the functions $f_{\mathbb{X}} : \mathbb{S} \to \mathbb{X}$ and $f_{\mathbb{Y}} : \mathbb{S} \to \mathbb{Y}$ and define the metrics as follows:

$$\textbf{Jaccard}(s) = \frac{|\operatorname{KNN}_k^{\mathbb{X}}(f_{\mathbb{X}}(s)) \cap \operatorname{KNN}_k^{\mathbb{Y}}(f_{\mathbb{X}}(s))|}{|\operatorname{KNN}_k^{\mathbb{X}}(f_{\mathbb{X}}(s)) \cup \operatorname{KNN}_k^{\mathbb{Y}}(f_{\mathbb{X}}(s))|}$$

$$\textbf{MRR}(s) = \frac{1}{\operatorname{Rank}_{\mathbb{Y}}(f_{\mathbb{X}}(s), f_{\mathbb{Y}}(s))}$$

$$\textbf{Cosine}(s) = \frac{f_{\mathbb{X}}(s) \cdot f_{\mathbb{Y}}(s)}{\|f_{\mathbb{X}}(s)\| \|f_{\mathbb{Y}}(s)\|}$$

where $\operatorname{KNN}_k^{\mathbb{A}}(\boldsymbol{v})$ is a function that returns the $k$-top similar samples (according to cosine similarity) to $\boldsymbol{v}$ in the space $\mathbb{A}$, and $\operatorname{Rank}_{\mathbb{A}}(\boldsymbol{v}, \boldsymbol{u})$ is a function that returns the index at which $\boldsymbol{u}$ is found in the ordered $\operatorname{KNN}_k^{\mathbb{A}}(\boldsymbol{v})$. The final score for each metric is the mean over each $s \in S$.

### A.5.2 RELATIVE REPRESENTATION SPACE CORRELATIONS

In this section, we analyze how similarities in absolute and relative spaces are correlated. Let us consider two spaces alignable in the relative space. We denote elements of the spaces with $\mathbb{A} \in \mathbb{R}^{m_1 \times n_1}$ and $\mathbb{B} \in \mathbb{R}^{m_2 \times n_2}$ and corresponding relative embeddings with $\mathbb{C} \in \mathbb{R}^{m_1 \times d}, \mathbb{D} \in \mathbb{R}^{m_2 \times d}$. Examples of $\mathbb{A}$ and $\mathbb{B}$ can be the `FastText` and `Word2Vec` word embedding spaces. We already observed in Table 1 how the spaces $\mathbb{A}$ and $\mathbb{B}$ are well aligned in the relative space. We can go further and analyze how self similarities in each space are preserved by the relative transform. In Figure 8, we show that relative representations not only facilitate latent space communication, but also preserve the underlying (absolute) latent space metric up to a certain degree.

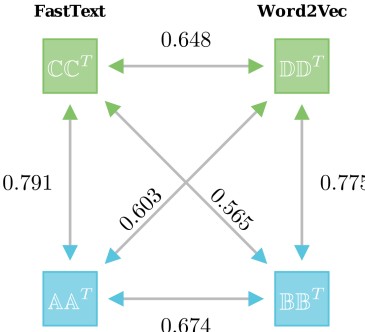

Figure 8: Self similiarities correlations between each space, measured with the Pearson correlation coefficient. In blue, we denote the self similarities in the absolute spaces $\mathbb{A}$, $\mathbb{B}$ of `FastText` and `Word2Vec`; in green we depict the relative spaces $\mathbb{C}$, $\mathbb{D}$. The correlation in the vertical arrows indicate how much the underlying metric in the abolute space is preserved by the relative coordinate transformation.

### A.5.3 LATENT DISTANCE AS A PERFORMANCE PROXY

The hyperperameters used in Section 4.2 are summarized in Table 10.

### A.5.4 TRAINING WITH ABSOLUTE VS. RELATIVE REPRESENTATIONS

The models trained on relative representations do not backpropagate through the anchors, which encourages a smoother optimization of the anchors' representations.

**Image Classification** The architecture is a standard deep CNN. We run a sweep for each dataset where we vary only the random seed (over 10 possible in total). We then aggregate by dataset and encoding type to obtain the final results with their standard deviation.

**Graph Classification**   We run a sweep identical to the one in Table 10 for the reference model, except that we sweep on the "Number of layers" with two values: 32 and 64. Each configuration is repeated with 10 different seeds, then we aggregate by dataset and encoding type to obtain the final results with their standard deviation.

### A.5.5   IMAGE RECONSTRUCTION

The relative and absolute models appearing in Figure 4 are vanilla AEs and VAEs, the same for all the datasets, and have a comparable number of trainable parameters. Their architecture is composed by simple convolutions, deconvolutions and mean squared error as reconstruction loss. The number of anchors is $500$ and the latent dimensionality of the absolute representations is $500$.

### A.5.6   TEXT CLASSIFICATION

We report in Tables 11 to 13 details on the transformers and anchors adopted in Section 5.2.

**Preprocessing**   Following the original work in which the `Amazon Reviews` dataset was proposed (Keung et al., 2020), we utilize both the *title* and *body* of each review. We differ in not using the category and in how we merge them; namely, we add the title as prefix for the body and add a full stop as separator when needed (avoiding duplicates). To obtain a single latent encoding for each sample, with fixed shape, we take the last hidden state and select the representation corresponding to the *[CLS]* token.

**Wikipedia anchors**   We use WikiMatrix, a corpus of sentences extracted from Wikipedia. The sentences are parallel between pairs of languages (i.e., same sentences translated in two languages), and since we are looking for a collection of parallel anchors between all 4 languages, we decided to use the English language as a pivot to compute the intersection. To get the final results, we considered only the sentences with margin score $\geq 1.06$, getting high-quality sentence alignments. In Table 13 we show the total number of parallel sentences when computing the intersections. We randomly selected 768 samples to use as anchors.

### A.5.7   IMAGE CLASSIFICATION

The details of the transformers used in Section 5.3 are summarized in Table 14.

### A.6   ADDITIONAL RESULTS

In this section we report additional results on the correlation between latent similarity and performance in Figure 9, results on the multilingual stitching both with Amazon coarse-grained in Table 15 and fine-grained in Table 16, results on the image classification stitching on `CIFAR-100` fine-grained in Table 18. Moreover, we evaluate the stitching performance of a multilingual transformer in Table 17.

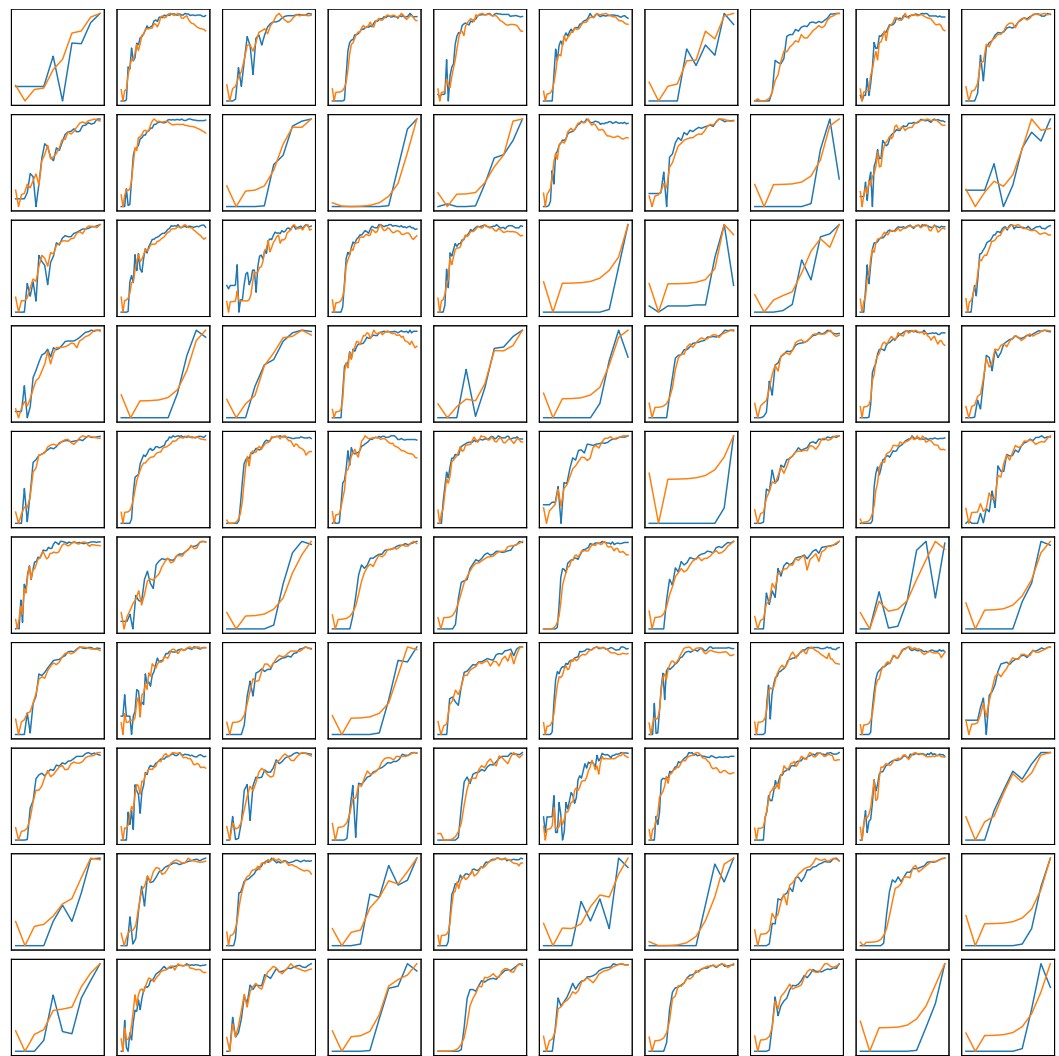

Figure 9: Correlation plot between performance and latent similarity with the reference model for multiple different models, over time.

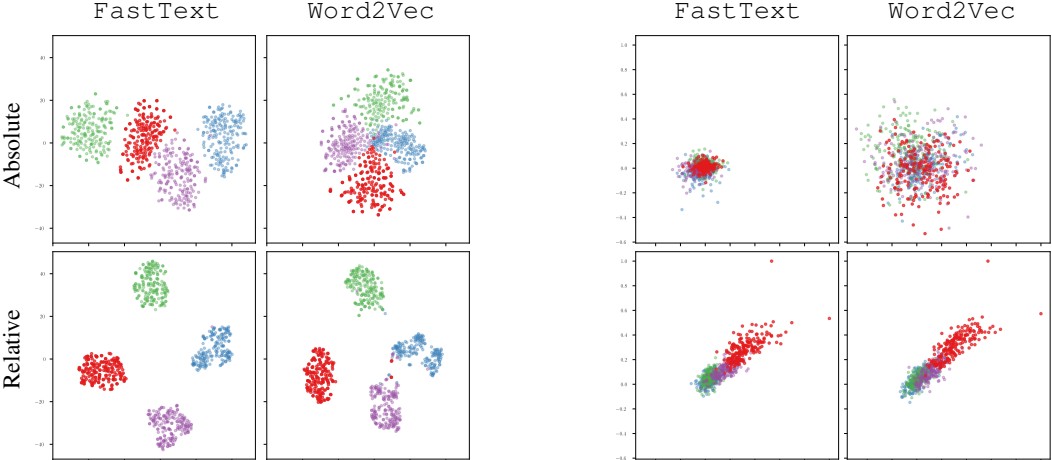

Figure 10: Same encodings as in Table 1 (left) but with tSNE *(left)* dimensionality reduction or visualizing only their first two dimensions *(right)*.

Table 7: Extended results from Section 4.1 with different anchor selection strategies. The table reports the mean score for each metric and its std across 10 different seeds.

| Mode | Type | Source | Target | Jaccard ↑ | MRR ↑ | Cosine ↑ |
|---|---|---|---|---|---|---|
| uniform | Absolute | FastText | FastText | $1.00 \pm 0.00$ | $1.00 \pm 0.00$ | $1.00 \pm 0.00$ |
| | | | Word2Vec | $0.00 \pm 0.00$ | $0.00 \pm 0.00$ | $0.01 \pm 0.00$ |
| | | Word2Vec | FastText | $0.00 \pm 0.00$ | $0.00 \pm 0.00$ | $0.01 \pm 0.00$ |
| | | | Word2Vec | $1.00 \pm 0.00$ | $1.00 \pm 0.00$ | $1.00 \pm 0.00$ |
| | Relative | FastText | FastText | $1.00 \pm 0.00$ | $1.00 \pm 0.00$ | $1.00 \pm 0.00$ |
| | | | Word2Vec | $0.34 \pm 0.01$ | $0.94 \pm 0.00$ | $0.86 \pm 0.00$ |
| | | Word2Vec | FastText | $0.39 \pm 0.00$ | $0.98 \pm 0.00$ | $0.86 \pm 0.00$ |
| | | | Word2Vec | $1.00 \pm 0.00$ | $1.00 \pm 0.00$ | $1.00 \pm 0.00$ |
| fps | Absolute | FastText | FastText | $1.00 \pm 0.00$ | $1.00 \pm 0.00$ | $1.00 \pm 0.00$ |
| | | | Word2Vec | $0.00 \pm 0.00$ | $0.00 \pm 0.00$ | $0.01 \pm 0.00$ |
| | | Word2Vec | FastText | $0.00 \pm 0.00$ | $0.00 \pm 0.00$ | $0.01 \pm 0.00$ |
| | | | Word2Vec | $1.00 \pm 0.00$ | $1.00 \pm 0.00$ | $1.00 \pm 0.00$ |
| | Relative | FastText | FastText | $1.00 \pm 0.00$ | $1.00 \pm 0.00$ | $1.00 \pm 0.00$ |
| | | | Word2Vec | $0.34 \pm 0.01$ | $0.94 \pm 0.00$ | $0.81 \pm 0.00$ |
| | | Word2Vec | FastText | $0.41 \pm 0.00$ | $0.98 \pm 0.00$ | $0.83 \pm 0.00$ |
| | | | Word2Vec | $1.00 \pm 0.00$ | $1.00 \pm 0.00$ | $1.00 \pm 0.00$ |
| kmeans | Absolute | FastText | FastText | $1.00 \pm 0.00$ | $1.00 \pm 0.00$ | $1.00 \pm 0.00$ |
| | | | Word2Vec | $0.00 \pm 0.00$ | $0.00 \pm 0.00$ | $0.01 \pm 0.00$ |
| | | Word2Vec | FastText | $0.00 \pm 0.00$ | $0.00 \pm 0.00$ | $0.01 \pm 0.00$ |
| | | | Word2Vec | $1.00 \pm 0.00$ | $1.00 \pm 0.00$ | $1.00 \pm 0.00$ |
| | Relative | FastText | FastText | $1.00 \pm 0.00$ | $1.00 \pm 0.00$ | $1.00 \pm 0.00$ |
| | | | Word2Vec | $0.35 \pm 0.00$ | $0.94 \pm 0.00$ | $0.87 \pm 0.00$ |
| | | Word2Vec | FastText | $0.39 \pm 0.00$ | $0.97 \pm 0.00$ | $0.87 \pm 0.00$ |
| | | | Word2Vec | $1.00 \pm 0.00$ | $1.00 \pm 0.00$ | $1.00 \pm 0.00$ |
| top1000 | Absolute | FastText | FastText | $1.00 \pm 0.00$ | $1.00 \pm 0.00$ | $1.00 \pm 0.00$ |
| | | | Word2Vec | $0.00 \pm 0.00$ | $0.00 \pm 0.00$ | $0.01 \pm 0.00$ |
| | | Word2Vec | FastText | $0.00 \pm 0.00$ | $0.00 \pm 0.00$ | $0.01 \pm 0.00$ |
| | | | Word2Vec | $1.00 \pm 0.00$ | $1.00 \pm 0.00$ | $1.00 \pm 0.00$ |
| | Relative | FastText | FastText | $1.00 \pm 0.00$ | $1.00 \pm 0.00$ | $1.00 \pm 0.00$ |
| | | | Word2Vec | $0.27 \pm 0.01$ | $0.84 \pm 0.01$ | $0.85 \pm 0.00$ |
| | | Word2Vec | FastText | $0.35 \pm 0.01$ | $0.97 \pm 0.00$ | $0.85 \pm 0.00$ |
| | | | Word2Vec | $1.00 \pm 0.00$ | $1.00 \pm 0.00$ | $1.00 \pm 0.00$ |
| top5000 | Absolute | FastText | FastText | $1.00 \pm 0.00$ | $1.00 \pm 0.00$ | $1.00 \pm 0.00$ |
| | | | Word2Vec | $0.00 \pm 0.00$ | $0.00 \pm 0.00$ | $0.01 \pm 0.00$ |
| | | Word2Vec | FastText | $0.00 \pm 0.00$ | $0.00 \pm 0.00$ | $0.01 \pm 0.00$ |
| | | | Word2Vec | $1.00 \pm 0.00$ | $1.00 \pm 0.00$ | $1.00 \pm 0.00$ |
| | Relative | FastText | FastText | $1.00 \pm 0.00$ | $1.00 \pm 0.00$ | $1.00 \pm 0.00$ |
| | | | Word2Vec | $0.32 \pm 0.00$ | $0.92 \pm 0.00$ | $0.86 \pm 0.00$ |
| | | Word2Vec | FastText | $0.38 \pm 0.00$ | $0.97 \pm 0.00$ | $0.86 \pm 0.00$ |
| | | | Word2Vec | $1.00 \pm 0.00$ | $1.00 \pm 0.00$ | $1.00 \pm 0.00$ |
| top10000 | Absolute | FastText | FastText | $1.00 \pm 0.00$ | $1.00 \pm 0.00$ | $1.00 \pm 0.00$ |
| | | | Word2Vec | $0.00 \pm 0.00$ | $0.00 \pm 0.00$ | $0.01 \pm 0.00$ |
| | | Word2Vec | FastText | $0.00 \pm 0.00$ | $0.00 \pm 0.00$ | $0.01 \pm 0.00$ |
| | | | Word2Vec | $1.00 \pm 0.00$ | $1.00 \pm 0.00$ | $1.00 \pm 0.00$ |
| | Relative | FastText | FastText | $1.00 \pm 0.00$ | $1.00 \pm 0.00$ | $1.00 \pm 0.00$ |
| | | | Word2Vec | $0.34 \pm 0.00$ | $0.93 \pm 0.00$ | $0.86 \pm 0.00$ |
| | | Word2Vec | FastText | $0.39 \pm 0.01$ | $0.97 \pm 0.00$ | $0.86 \pm 0.00$ |
| | | | Word2Vec | $1.00 \pm 0.00$ | $1.00 \pm 0.00$ | $1.00 \pm 0.00$ |

Table 8: Generalization of the results from Section 4.1 on word embeddings to a different data modality, with different anchor selection strategies (See Appendix A.2 for their description). The dataset considered is CIFAR-10, and the table reports the mean score for each metric and its std across 10 different seeds.

| Mode | Type | Source | Target | Jaccard ↑ | MRR ↑ | Cosine ↑ |
|---|---|---|---|---|---|---|
| uniform | Absolute | ViT-base | ViT-base | $1.00 \pm 0.00$ | $1.00 \pm 0.00$ | $1.00 \pm 0.00$ |
| | | | ViT-small | - | - | - |
| | | ViT-small | ViT-base | - | - | - |
| | | | ViT-small | $1.00 \pm 0.00$ | $1.00 \pm 0.00$ | $1.00 \pm 0.00$ |
| | Relative | ViT-base | ViT-base | $1.00 \pm 0.00$ | $1.00 \pm 0.00$ | $1.00 \pm 0.00$ |
| | | | ViT-small | $0.11 \pm 0.00$ | $0.27 \pm 0.01$ | $0.97 \pm 0.00$ |
| | | ViT-small | ViT-base | $0.11 \pm 0.00$ | $0.30 \pm 0.01$ | $0.97 \pm 0.00$ |
| | | | ViT-small | $1.00 \pm 0.00$ | $1.00 \pm 0.00$ | $1.00 \pm 0.00$ |
| fps | Absolute | ViT-base | ViT-base | $1.00 \pm 0.00$ | $1.00 \pm 0.00$ | $1.00 \pm 0.00$ |
| | | | ViT-small | - | - | - |
| | | ViT-small | ViT-base | - | - | - |
| | | | ViT-small | $1.00 \pm 0.00$ | $1.00 \pm 0.00$ | $1.00 \pm 0.00$ |
| | Relative | ViT-base | ViT-base | $1.00 \pm 0.00$ | $1.00 \pm 0.00$ | $1.00 \pm 0.00$ |
| | | | ViT-small | $0.12 \pm 0.00$ | $0.37 \pm 0.01$ | $0.96 \pm 0.00$ |
| | | ViT-small | ViT-base | $0.12 \pm 0.00$ | $0.39 \pm 0.01$ | $0.96 \pm 0.00$ |
| | | | ViT-small | $1.00 \pm 0.00$ | $1.00 \pm 0.00$ | $1.00 \pm 0.00$ |
| kmeans | Absolute | ViT-base | ViT-base | $1.00 \pm 0.00$ | $1.00 \pm 0.00$ | $1.00 \pm 0.00$ |
| | | | ViT-small | - | - | - |
| | | ViT-small | ViT-base | - | - | - |
| | | | ViT-small | $1.00 \pm 0.00$ | $1.00 \pm 0.00$ | $1.00 \pm 0.00$ |
| | Relative | ViT-base | ViT-base | $1.00 \pm 0.00$ | $1.00 \pm 0.00$ | $1.00 \pm 0.00$ |
| | | | ViT-small | $0.11 \pm 0.00$ | $0.25 \pm 0.01$ | $0.97 \pm 0.00$ |
| | | ViT-small | ViT-base | $0.10 \pm 0.00$ | $0.27 \pm 0.00$ | $0.97 \pm 0.00$ |
| | | | ViT-small | $1.00 \pm 0.00$ | $1.00 \pm 0.00$ | $1.00 \pm 0.00$ |

Table 9: All the datasets utilized in our work with their number of classes.

| | Dataset | Number of Classes |
|---|---|---|
| Image | MNIST | 10 |
| | Fashion MNIST | 10 |
| | CIFAR-10 | 10 |
| | CIFAR-100 | 20 (coarse) — 100 (fine) |
| | ImageNet1k | 1000 |
| Graph | Cora | 7 |
| | CiteSeer | 6 |
| | PubMed | 3 |
| Text | TREC | 6 (coarse) — 50 (fine) |
| | DBpedia | 14 |
| | Amazon Reviews | 2 (coarse) — 5 (fine) |

Table 10: The reference model and exhaustive hyperparameter combinations pertaining Section 4.2.

| Hyperparameter | Reference Model | Sweep |
|---|---|---|
| Seed | 1 | 0, 1, 2, 3, 4 |
| Epochs | 500 | 10, 30, 50 |
| Number of layers | 32 | 32, 64 |
| Dropout Probability | 0.5 | 0.1, 0.5 |
| Hidden Activations | ReLU | ReLU, Tanh |
| Convolution Activation | ReLU | ReLU, Tanh |
| Optimizer | Adam | Adam, SGD |
| Learning Rate | 0.02 | 0.01, 0.02 |
| Graph Embedder | GCNConv | GCNConv, GINConv |

Table 11: The HuggingFace transformers employed in Section 5.2 to tackle the *Cross-lingual* setting.

| Language | HuggingFace transformers name | Encoding Dim |
|---|---|---|
| English | roberta-base | 768 |
| Spanish | PlanTL-GOB-ES/roberta-base-bne | 768 |
| French | ClassCat/roberta-base-french | 768 |
| Japanese | nlp-waseda/roberta-base-japanese | 768 |

Table 12: The HuggingFace transformers employed in Section 5.2 to tackle the *Cross-architecture* setting.

| HuggingFace transformers name | Encoding Dim |
|---|---|
| bert-base-cased | 768 |
| bert-base-uncased | 768 |
| google/electra-base-discriminator | 768 |
| roberta-base | 768 |

Table 13: WikiMatrix analysis. Each row shows the number of parallel sentences having a translation available in all the languages of that row. Since we consider all four languages, we have 3338 parallel sentences available.

| Languages | Number of Sentences |
|---|---|
| en, es | 2302527 |
| en, ja | 264259 |
| en, fr | 1682477 |
| en, es, fr | 23200 |
| en, es, ja | 147665 |
| en, fr, ja | 20990 |
| en, es, fr, ja | **3338** |

Table 14: Timm transformers used in Section 5.3.

| Version | Timm model name | Encoding Dim | Training data |
|---|---|---|---|
| ViT | vit_base_patch16_224 | 768 | JFT-300M, ImageNet |
| ViT | vit_small_patch16_224 | 384 | ImageNet |
| ViT | vit_base_resnet50_384 | 768 | ImageNet |
| RexNet | rexnet_100 | 1280 | ImageNet |

Table 15: Stitching performance comparison with different encodings techniques. The table reports the mean weighted F1 (± std) and MAE classification performance on `Amazon Reviews` coarse-grained, across 5 different seeds. All the language pairs are shown.

| | | Absolute | | Relative | | | |
| | | | | Translated | | Wikipedia | |
| Decoder | Encoder | FScore | MAE | FScore | MAE | FScore | MAE |
|---|---|---|---|---|---|---|---|
| en | en | 91.54 ± 0.58 | 0.08 ± 0.01 | 90.06 ± 0.60 | 0.10 ± 0.01 | 90.45 ± 0.52 | 0.10 ± 0.01 |
| | es | 43.67 ± 1.09 | 0.56 ± 0.01 | 82.78 ± 0.81 | 0.17 ± 0.01 | 78.53 ± 0.30 | 0.21 ± 0.00 |
| | fr | 54.41 ± 1.61 | 0.45 ± 0.02 | 78.49 ± 0.66 | 0.21 ± 0.01 | 70.41 ± 0.57 | 0.29 ± 0.01 |
| | ja | 48.72 ± 0.90 | 0.51 ± 0.01 | 65.72 ± 0.55 | 0.34 ± 0.01 | 66.31 ± 0.80 | 0.34 ± 0.01 |
| es | en | 33.23 ± 1.00 | 0.66 ± 0.01 | 78.68 ± 2.74 | 0.21 ± 0.03 | 76.65 ± 3.23 | 0.23 ± 0.03 |
| | es | 91.64 ± 1.02 | 0.08 ± 0.01 | 89.96 ± 0.77 | 0.10 ± 0.01 | 89.62 ± 0.94 | 0.10 ± 0.01 |
| | fr | 47.66 ± 0.70 | 0.52 ± 0.01 | 78.57 ± 1.80 | 0.21 ± 0.02 | 75.25 ± 0.76 | 0.25 ± 0.01 |
| | ja | 53.10 ± 2.27 | 0.46 ± 0.02 | 67.69 ± 0.24 | 0.32 ± 0.00 | 61.84 ± 0.61 | 0.38 ± 0.01 |
| fr | en | 51.00 ± 2.63 | 0.49 ± 0.03 | 83.32 ± 1.80 | 0.17 ± 0.02 | 75.55 ± 0.37 | 0.24 ± 0.00 |
| | es | 51.96 ± 2.81 | 0.48 ± 0.03 | 82.50 ± 0.83 | 0.17 ± 0.01 | 77.12 ± 0.88 | 0.23 ± 0.01 |
| | fr | 88.22 ± 0.75 | 0.12 ± 0.01 | 85.68 ± 1.37 | 0.14 ± 0.01 | 86.45 ± 0.96 | 0.13 ± 0.01 |
| | ja | 50.32 ± 4.16 | 0.50 ± 0.04 | 69.38 ± 0.73 | 0.31 ± 0.01 | 62.79 ± 0.27 | 0.37 ± 0.00 |
| ja | en | 53.82 ± 2.62 | 0.46 ± 0.03 | 68.66 ± 3.62 | 0.31 ± 0.04 | 70.26 ± 3.16 | 0.29 ± 0.03 |
| | es | 44.91 ± 2.21 | 0.55 ± 0.02 | 70.37 ± 6.94 | 0.29 ± 0.06 | 58.54 ± 1.21 | 0.41 ± 0.01 |
| | fr | 66.46 ± 1.30 | 0.34 ± 0.01 | 76.49 ± 1.13 | 0.23 ± 0.01 | 63.94 ± 2.70 | 0.36 ± 0.02 |
| | ja | 83.30 ± 0.67 | 0.17 ± 0.01 | 81.04 ± 0.82 | 0.19 ± 0.01 | 80.80 ± 1.25 | 0.19 ± 0.01 |

Table 16: Stitching performance comparison with different encodings techniques. The table reports the mean weighted F1 (± std) and MAE classification performance on `Amazon Reviews` fine-grained, across 5 different seeds. All the language pairs are shown.

| | | Absolute | | Relative | | | |
| | | | | Translated | | Wikipedia | |
| Decoder | Encoder | FScore | MAE | FScore | MAE | FScore | MAE |
|---|---|---|---|---|---|---|---|
| en | en | 65.46 ± 2.89 | 0.38 ± 0.02 | 61.18 ± 1.92 | 0.44 ± 0.02 | 62.36 ± 2.23 | 0.43 ± 0.02 |
| | es | 22.70 ± 0.41 | 1.39 ± 0.03 | 51.67 ± 1.20 | 0.62 ± 0.01 | 45.40 ± 0.68 | 0.76 ± 0.01 |
| | fr | 30.75 ± 0.67 | 1.19 ± 0.02 | 49.18 ± 0.83 | 0.69 ± 0.02 | 40.29 ± 0.90 | 0.91 ± 0.02 |
| | ja | 24.85 ± 0.91 | 1.37 ± 0.07 | 37.34 ± 1.49 | 0.99 ± 0.02 | 37.73 ± 0.70 | 1.01 ± 0.02 |
| es | en | 21.24 ± 0.81 | 1.43 ± 0.07 | 51.02 ± 2.54 | 0.68 ± 0.05 | 47.70 ± 5.08 | 0.73 ± 0.10 |
| | es | 61.29 ± 3.04 | 0.43 ± 0.02 | 57.89 ± 3.80 | 0.48 ± 0.03 | 57.96 ± 4.40 | 0.48 ± 0.03 |
| | fr | 29.02 ± 0.85 | 1.26 ± 0.05 | 48.40 ± 1.02 | 0.71 ± 0.02 | 44.92 ± 1.83 | 0.77 ± 0.01 |
| | ja | 29.23 ± 1.32 | 1.22 ± 0.02 | 37.22 ± 1.56 | 1.03 ± 0.04 | 34.56 ± 0.87 | 1.08 ± 0.04 |
| fr | en | 27.39 ± 1.22 | 1.23 ± 0.06 | 45.55 ± 3.55 | 0.76 ± 0.09 | 39.01 ± 1.25 | 0.88 ± 0.06 |
| | es | 29.47 ± 3.68 | 1.18 ± 0.07 | 40.29 ± 1.72 | 0.90 ± 0.04 | 41.29 ± 2.01 | 0.83 ± 0.04 |
| | fr | 56.40 ± 1.89 | 0.51 ± 0.01 | 53.58 ± 0.70 | 0.57 ± 0.01 | 54.23 ± 0.95 | 0.56 ± 0.01 |
| | ja | 25.92 ± 1.31 | 1.25 ± 0.05 | 38.60 ± 1.03 | 0.96 ± 0.02 | 35.22 ± 0.56 | 1.08 ± 0.02 |
| ja | en | 29.36 ± 0.59 | 1.17 ± 0.04 | 38.19 ± 2.28 | 0.88 ± 0.03 | 36.57 ± 1.72 | 0.98 ± 0.02 |
| | es | 25.64 ± 1.77 | 1.28 ± 0.04 | 34.23 ± 2.62 | 1.00 ± 0.05 | 33.16 ± 2.28 | 1.06 ± 0.03 |
| | fr | 31.79 ± 1.91 | 1.06 ± 0.02 | 38.50 ± 2.46 | 0.89 ± 0.02 | 36.68 ± 3.14 | 1.00 ± 0.05 |
| | ja | 54.09 ± 1.35 | 0.60 ± 0.02 | 50.89 ± 1.70 | 0.65 ± 0.02 | 51.64 ± 1.47 | 0.65 ± 0.02 |

Table 17: Stitching performance comparison on XLM-R, a multilingual model by design. The table reports the mean weighted F1 (± std) and MAE classification performance on `Amazon Reviews` fine-grained, across 5 different seeds.

| Decoder | Encoder | Absolute | | Relative | |
|---|---|---|---|---|---|
| | | FScore | MAE | FScore | MAE |
| en | en | $65.27 \pm 0.94$ | $0.41 \pm 0.01$ | $58.24 \pm 1.92$ | $0.51 \pm 0.03$ |
| | es | $59.55 \pm 0.76$ | $0.48 \pm 0.01$ | $52.81 \pm 1.57$ | $0.62 \pm 0.02$ |
| | fr | $58.58 \pm 1.04$ | $0.49 \pm 0.01$ | $54.01 \pm 1.34$ | $0.59 \pm 0.02$ |
| | ja | $57.98 \pm 0.77$ | $0.52 \pm 0.01$ | $48.47 \pm 2.67$ | $0.71 \pm 0.04$ |
| es | en | $60.32 \pm 1.50$ | $0.47 \pm 0.01$ | $45.69 \pm 2.19$ | $0.87 \pm 0.07$ |
| | es | $61.25 \pm 1.74$ | $0.44 \pm 0.01$ | $57.61 \pm 0.73$ | $0.51 \pm 0.01$ |
| | fr | $59.50 \pm 1.41$ | $0.47 \pm 0.01$ | $45.16 \pm 3.30$ | $0.83 \pm 0.09$ |
| | ja | $58.24 \pm 1.31$ | $0.51 \pm 0.02$ | $41.14 \pm 1.76$ | $0.99 \pm 0.05$ |
| fr | en | $58.00 \pm 4.21$ | $0.49 \pm 0.03$ | $52.37 \pm 1.66$ | $0.66 \pm 0.03$ |
| | es | $56.87 \pm 3.79$ | $0.49 \pm 0.03$ | $54.99 \pm 0.46$ | $0.57 \pm 0.01$ |
| | fr | $57.99 \pm 3.88$ | $0.47 \pm 0.02$ | $57.00 \pm 0.90$ | $0.52 \pm 0.01$ |
| | ja | $55.83 \pm 3.32$ | $0.53 \pm 0.03$ | $39.15 \pm 1.21$ | $1.02 \pm 0.03$ |
| ja | en | $59.53 \pm 1.73$ | $0.48 \pm 0.01$ | $39.46 \pm 2.34$ | $1.04 \pm 0.07$ |
| | es | $57.02 \pm 1.36$ | $0.51 \pm 0.00$ | $40.74 \pm 2.75$ | $0.97 \pm 0.09$ |
| | fr | $57.48 \pm 1.06$ | $0.51 \pm 0.01$ | $43.36 \pm 3.70$ | $0.89 \pm 0.11$ |
| | ja | $61.43 \pm 0.97$ | $0.45 \pm 0.01$ | $57.67 \pm 1.17$ | $0.51 \pm 0.01$ |

Table 18: Stitching performance comparison with different encodings techniques. The table reports the mean weighted F1 (± std) classification performance on `CIFAR-100` fine-grained, across 5 different seeds.

| Decoder | Encoder | Absolute | Relative |
|---|---|---|---|
| rexnet-100 | rexnet-100 | $72.77 \pm 0.19$ | $71.39 \pm 0.18$ |
| | vit-base-patch16-224 | - | $40.68 \pm 0.50$ |
| | vit-base-resnet50-384 | - | $38.18 \pm 0.24$ |
| | vit-small-patch16-224 | - | $44.11 \pm 0.84$ |
| vit-base-patch16-224 | rexnet-100 | - | $57.81 \pm 0.39$ |
| | vit-base-patch16-224 | $88.69 \pm 0.14$ | $87.05 \pm 0.34$ |
| | vit-base-resnet50-384 | $1.08 \pm 0.19$ | $66.65 \pm 1.79$ |
| | vit-small-patch16-224 | - | $73.73 \pm 0.60$ |
| vit-base-resnet50-384 | rexnet-100 | - | $66.91 \pm 0.79$ |
| | vit-base-patch16-224 | $1.10 \pm 0.09$ | $75.70 \pm 0.68$ |
| | vit-base-resnet50-384 | $85.85 \pm 0.18$ | $85.04 \pm 0.38$ |
| | vit-small-patch16-224 | - | $75.52 \pm 0.36$ |
| vit-small-patch16-224 | rexnet-100 | - | $56.60 \pm 0.39$ |
| | vit-base-patch16-224 | - | $70.14 \pm 0.46$ |
| | vit-base-resnet50-384 | - | $62.85 \pm 1.22$ |
| | vit-small-patch16-224 | $84.11 \pm 0.14$ | $83.24 \pm 0.13$ |

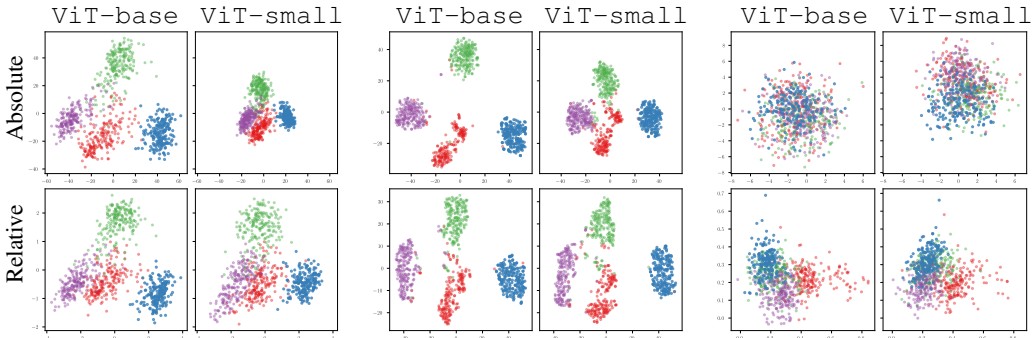

Figure 11: Different dimensionality reduction techniques applied to absolute and relative spaces on `CIFAR-10`. From left to right: PCA (Principal Component Analysis), tSNE, and visualizing only their first two dimensions. Only 800 randomly sampled points are shown, belonging to the classes "bird", "ship", "cat", and "frog".

