# OpenReview forum: "Relative representations enable zero-shot latent space communication"
_ICLR.cc/2023/Conference — ICLR 2023 notable top 5%_

### Official Review · Reviewer_LWWZ · 2022-10-27

**Confidence:** 4
**Correctness:** 4
**Technical Novelty And Significance:** 4
**Empirical Novelty And Significance:** 4
**Recommendation:** 10

**Clarity, Quality, Novelty And Reproducibility:**

- Clarity: the paper and its experiments are very clearly motivated and presented
- Quality: the paper is very plausibly motivated and its main hypothesis is solidly supported by thorough empirical demonstrations
- Novelty: the proposed idea and applications are very novel
- Reproducibility: the submission includes code and visualization scripts to reproduce the results reported in the paper

**Strength And Weaknesses:**

Strengths:
- Simple and at the same time very innovative idea. Its simplicity makes it extremely relevant and applicable across settings, domains, architectures and training paradigms, suggesting great potential for impact, both conceptual as well as practical.
- Thorough examination and empirical demonstration of some of the potential of this new technique.

The paper does not have any apparent major weaknesses of relevance.
There are however technical details that could benefit from some clarification. For instance, the paper considers the training modality of training models with relative representation, but while doing that it is not readily clear whether this happens by also backpropagating through the anchors. It would be beneficial to clarify that, and whether and if there are any differences between doing that (backpropagating also through the anchors) vs only backpropagating through the data.

**Summary Of The Paper:**

The paper introduces the concept and the use of relative representations for machine learning models, which consists in representing data samples in terms of their embedding distance to a fixed set of "anchor data samples". The motivation behind proposing this representation is that, under some regularity and symmetry assumptions, it would allow for representing data in a way that is invariant across training run, initializations and even architectures. In fact, the paper demonstrates in simulations on various datasets and tasks that this hypothesis is empirically borne out. Interestingly, data similarity in this representation is observed to be highly correlated with model performance. A tantalizing application of the invariance afforded by relative representations is that they allow for zero-shot model-stitching, i.e. "transferring" representations across models, possibly trained on different datasets or with differing architectures. The paper for instance demonstrates combining together encoder and decoder models that were not originally trained together, which allow for instance for things like cross-lingual stitching: combining encoders and decoders trained on different languages.

**Summary Of The Review:**

The paper introduces the concept of relative representations, a simple idea with straightforward realization but far reaching conceptual and practical consequences. The applications that this conceptual innovation affords are wide reaching and include potential theoretical advances in the study of generalization and invariance in representation learning, and a host of practical applications such as zero-shot stitching of models.

---

> ### Author Response · Authors · 2022-11-16
> **Response to Reviewer LWWZ**
>
> We want to thank the reviewer for appreciating our work and strongly pushing for its acceptance. We agree that the reuse of the models and computational resources spent is extremely important, favoring an energy efficiency approach currently lacking in the AI field.
>
> We confirm the reviewer's intuition on backpropagation through the anchors: the gradient does not flow through them. When training the encoder, the training shows more instabilities if we do not *stop-grad the anchor's gradients*. We believe this happens because the anchors change more abruptly, leading to a rapid-changing relative representation. We added this implementation detail in the appendix.
>
> We hope that our reply clarifies the relative training procedure. However, we are open to further discussion.

---

> > ### Comment · Reviewer_LWWZ · 2022-11-18
> > **Acknowledgement of response from the authors**
> >
> > I want to thank the authors for their clarifications that satisfactorily address my questions.

---

### Official Review · Reviewer_hU9r · 2022-10-27

**Confidence:** 4
**Correctness:** 3
**Technical Novelty And Significance:** 3
**Empirical Novelty And Significance:** 3
**Recommendation:** 8

**Clarity, Quality, Novelty And Reproducibility:**

I have no concerns about clarity, quality, or reproducibility. The ideas were clearly presented, I found no mathematical errors, and the authors include a thorough appendix of specific hyperparameters used along with supplemental code.

I am less confident about evaluating the novelty. The idea is straightforward and simple enough that I was surprised that it has not been tried before. In section 3.1, the authors present the idea using the more abstract notion of "a vector of similarities", but ultimately specialize to simply using the dot product of normalized vectors. Ultimately, as used by the authors, the idea is simply:

> 1. Identify $k$ "anchor" elements from your inputs, and let $A$ be the normalized vector representation of these anchor elements.
> 2. Let $P$ be a linear transformation which takes the vectors of $A$ to the standard basis.
> 3. To form the "relative representation" of a given vector representation of any input, we simply normalize and apply $P$.

The authors then emphasize that this relative representation is, by construction, invariant to a variety of transformations, and can be reliably used to align the latent spaces of different models, assuming they both have latent vector representations for the anchor elements (or, as in the case of machine translation, "parallel anchors"). I was unable to uncover any prior work during a literature search.

**Strength And Weaknesses:**

### Strengths
1. The fundamental idea is pleasingly simple
2. The authors included experiments to validate their motivation and intuition
3. A comprehensive suite of empirical results on the ultimate task (stitching), including multiple datasets, domains, and variations (eg. differences in architecture vs. data) were included

### Weaknesses
1. For the main tasks (stitching) presented in section 5 I do not think the authors made it sufficiently clear which models were *trained* using relative embeddings, and which models used the relative embeddings purely as a post-hoc adjustment step. The emphasis on zero-shot (in the title, section header, and throughout), and the fact that earlier experiments (eg. section 4.1) were performed by creating the relative embeddings post-hoc may leave the reader with the impression that the relative embeddings can allow existing pre-trained networks to be stitched together. To my understanding, this is fundamentally impossible. At a minimum, the parts of the model downstream of the "stiching" must always be trained using relative embeddings, because the function which takes an absolute embedding to a relative one is not (in general) invertible. Of course, this also opens two opportunities for future work: (a) For instances where the mapping from absolute to relative embeddings is not strictly invertible, perhaps it is still essentially invertible on the data manifold. For example, a large component which contributes to the lack of invertibility is the vector normalization operation, however it has previously been argued that regularization results in embeddings which are essentially on a sphere. In such a setting, we could "learn" the inverse via gradient descent, and attempt to use this to stitch together networks without fine-tuning either part. (b) Can we choose the anchor nodes to improve the extent to which the transformation is essentially invertible?

2. I have a few issues with Section 4.1 - Word Embeddings. First, using 300 randomly drawn parallel anchors seems very weak - if the words were drawn uniformly randomly then they are very likely all rare words, and as such would seem to serve as a very poor choice for anchors. This also may explain why the Jaccard similarity did not increase as substantially as might have been expected. I also take issue with this sentence: "The average Jaccard distance reported in Table 1 (right), says that the neighborhoods of the relative representations are matched exactly 34% of the time in one direction, and 39% of the time in the other one." At least to my understanding, that is not what the average of Jaccard similarity would show. In fact, one could obtain those metrics with neighborhoods which *never* exactly match. Perhaps what was meant is that the *words* in the neighborhoods matched exactly, as the authors then go on to mention the discrepancies are likely due to semantic differences. I agree, and as such I wonder why such a course-grained evaluation was used in the first place? The setting we are in is as follows: given some embeddings A and B, find some new embeddings C and D such that the all-pairs similarities between elements of A are proportional to those between elements of C (and similarly with B and D), however the similarities between equal elements of C and D are minimized. One could simply measure and report on that metric.

3. No theoretical analysis is included. It is, of course, increasingly common for papers to rely fundamentally on empirical results, however it seems possible to prove results relating the number of anchor entities, variation in the training data, and potential accuracy on the task. Even making minor preliminary theoretical statements on these aspects would strengthen the paper.

4. I was not able to find details on how the 2-dimensional representation in on the left half of Table 1 was created. Presumably, some dimensionality reduction technique (eg. tSNE) is used, however that raises into question the usefulness of drawing conclusions from such a picture as the dimensionality reduction introduces a number of factors which may impact the resulting representation. (One could imagine various dimensionality reduction techniques with noise or hyperparameters which, for the same exact embedding, result in different representations.) At the very least, the authors should explain how these pictures were derived.


### Typos / Suggestions
1. Abstract: I would recommend including something about the anchor entities in the abstract, as (to my mind) that is essential to understanding the high-level idea. Perhaps this sentence: "In this work, we propose to adopt pairwise similarities as an alternative data representation, that can be used to enforce the desired invariance without any additional training." could be changed to: "In this work, we propose to use the similarity between each representation and a fixed set of anchor representations, and demonstrate that this can enforce the desired invariances without any additional training."
2. Page 1: "The underlying assumption is that the learned latent spaces should be the best encoding given the data distribution, the downstream task, and the network constraints. In practice, however, the learned latent spaces are subject to changes even when the above assumptions remain fixed." The writing here presents this as though this is contradictory, but of course it is not - the latent representation which is best for the downstream task given network constraints is not unique.
3. Page 2: "more in general" -> "more generally"
4. Page 7, Figure 4: I don't think this figure highlights the results as well as it could. Consider making the figure 5 rows tall, and group based on absolute vs. relative first (i.e. rows would be original, abs ae, abs vae, rel ae, rel vae). This would make it very clear that absolute struggles, while relative works well. (I would make similar suggestions for Table 3 and 5)
5. Page 7: "prove that representations are invariant to training stochasticity"is a bit too strong, as I believe any mathematically rigorous interpretation of this statement would be provably false. Something like "support our claim that relative representations are more robust to training stochasticity" seems more accurate, given the results presented.
6. Page 8: "obtaine" -> "obtaining"
7. Page 9: "allow to stitch modules" -> "allow stitching modules"

**Summary Of The Paper:**

This paper highlights the difficulty in using latent representations which were obtained by training on different data, with different architectures, or, indeed, simply as a consequence of stochasticity in the training process itself. They introduce the notion of a "relative embedding", which is obtained by projecting the latent representations onto a (fixed, shared) set of representations of "anchor entities". The authors provide empirical evidence for the validity of their approach in several settings:
1. Aligning latent representations allows for greater preservation of similarity in a shallow word embedding setting.
2. Using relative embeddings during training does not significantly decrease performance on a collection of image and graph node classification tasks.
3. Relative embeddings allow for zero-shot model stitching, wherein models with different architectures and potentially trained on different tasks can be combined when using these relative embeddings.


**Summary Of The Review:**

This paper presents a simple idea that seems to work very well. The practical limitation of the proposed method is that it only works if the networks were trained with these relative representations in the first place. The strongest direct outcome of this paper would be that people would start using relative representations by default, thus facilitating the "stitching" mentioned in section 5, however I think this would require much greater exploration (eg. heuristic methods to choose the anchor elements with theoretical bounds on the potential impact) before being widely adopted.

---

> ### Author Response · Authors · 2022-11-16
> **Response to Reviewer hU9r (1/2)**
>
> We thank the reviewer for the insightful review and comments. We would like to address the following concerns:
>
> 1. We further clarified which experiments are trained on relative representations, highlighting that all the zero-shot stitching experiments require a decoder trained on relative representations. We include here a short list for clarity: Section 4.3 (absolute vs relative training procedures), Sections 5.* (zero-shot stitching experiments).
> Our intention in this work was to emphasize the zero-shot capabilities for “latent communication” (e.g., comparing latent spaces), where by “zero-shot” we indicate the absence of any retraining procedure after stitching together pre-trained components. We kindly ask the reviewer to view our stitching experiments primarily in the light of a methodology to analyze latent communication rather than the main application. We are convinced future works will build upon the knowledge gained from these experiments to obtain substantial practical applications.
>
>     We are glad that the reviewer shares our interest in the practical **inverse problem of relative-to-absolute transformation**. As a matter of fact, we are already exploring it as a possible follow-up with a significant impact (more details in the “Basis change” and “Theoretical Analysis” paragraphs): employing the relative space to communicate with any pre-trained absolute neural component.
>
> 2. We thank the reviewer for the improvement suggestions in Section 4.1. Following those advices:
>
>     a. **We repeated the experiment 10 times** with different random anchors and report mean and standard deviation.
>
>     b. Indeed our metrics are discrete and consider word neighborhoods; we clarified their explanation in Section 4.1 and Appendix A.5.1.
>
>     c. We **added a continuous metric** (Cosine) to compare the latent spaces directly.
>
>     d. We are not sure we correctly understood the proposed metric and would like clarification on its definition. From our understanding, the
>          setting described:
>
>       > given some embeddings A (*absolute*) and B (*absolute*), find some new embeddings C (*relative*) and D (*relative*) such that the all-pairs similarities between elements of A (*absolute*) are proportional to those between elements of C (*relative*)”
>
>      seems to compare an absolute space and its corresponding relative space (A-C and B-D).  Although interesting, this differs from what we show in Section 4.1 and generally in the paper, where we compare different spaces of the same “type” (absolute/relative, i.e., A-B and C-D).  We kindly ask the reviewer to clarify this metric, which we could add in the appendix as an extra experiment, or confirm it to be equivalent to the Cosine similarity we already added as part of Table 1 in Section 4.1.
>
>     e. We confirm that the anchor selection strategy is significant and it is possible to improve over random selection. Please see the added analysis in Table 7 and Table 8. However, this work focuses on introducing the relative representations and showing that a **random selection strategy is enough to achieve latent communication**. We are sure that future works will improve the performance.
>
>     f. Please also refer to the [answer to *Reviewer LNFQ*](https://openreview.net/forum?id=SrC-nwieGJ&noteId=w4rCQ-xjhn) for other improvements to Section 4.1, particularly Table 8 and Figure 10, which introduce the same experiment on a new data modality (images).
>
> 3. In the left half of Table 1, the PCA dimensionality reduction was used to visualize the embeddings. We agree it is important to avoid visualization artifacts when visually analyzing latent spaces. We improved Table 1 by:
>
>     a. **Specifying the dimensionality reduction employed in its caption**
>
>     b. **Adding equivalent Figures in the appendix with other dimensionality reductions, see Figure 9** (i.e., TSNe and only the first two dimensions without any dimensionality reduction)
>
> 4. To the best of our efforts, we could not uncover prior works during literature searches and discussions with prominent researchers in the field (who we plan to acknowledge in the camera ready). Surprisingly, too little effort has been made to study how different neural networks can interact. We hope that this work can promote this research direction.
>
> 5. We thank the reviewer for pointing out a better pitch for the abstract. We added a brief mention about the anchors and their role, hoping this would help the reader and enhance the clarity.
>
> *(continued)*

---

> > ### Author Response · Authors · 2022-11-16
> > **Response to Reviewer hU9r (2/2)**
> >
> >
> > ### Basis change
> > Our method is not a basis change since anchors are not guaranteed to be orthogonal. We are investigating a follow-up direction where we explore a “basis change” approach instead of a normalized projection (cosine similarity). We show preliminary experimental evidence in this direction: in [this figure](https://i.ibb.co/wNtzkq8/basis-change.png), a synthetic relative representation with all zeros but in one dimension is used to reconstruct MNIST digits. The relative representation obtained with a basis change (top row) reflects the movement along the anchor dimension; meanwhile, the inner product does not attach any semantic to the latent dimension (bottom row).
> >
> > ### Theoretical analysis
> > We agree with the reviewer’s comments on the importance of including a theoretical analysis of our method, and we are currently investigating it as a follow-up direction. We discovered that the fundamental empirical assumption that latent spaces differ by an isometric transformation on which our work builds on can be related to theoretical results on linear identifiability of deep neural models in the nonlinear Independent Component Analysis (ICA) setting ([1],[2],[3],[4]). This line of works serves as a solid theoretical justification for papers on stitching (Lenc & Vedaldi (2015); Bansal et al. (2021); Csiszárik et al. (2021)), which basically assumes that different latent spaces differ by a linear transformation and train a linear layer to connect latent spaces. Our setting is stricter as we assume the linear transformation is indeed an isometry. We are planning to extend the analysis of ([1], [2]) to our isometric setting.
> >
> > The number of anchors and their selection strategy are worthy of a separate in-depth investigation (as preliminary shown empirically by Figure 6, Table 8 and 9). We are optimistic that a formal characterization of the relation between this choice and some performance metric on the task is possible by analyzing the invertibility,  or a weaker notion of it, of the transformation matrix between the absolute to the relative spaces. This enables quantification of the information loss when changing the representation ( e.g., full invertibility ensures zero information loss) and therefore a lower bound on the maximum downstream performance attainable maintaining the isometry invariance through relative representation. For example, consider the particular case where the anchors coincide with the entire dataset, and the similarity function employed is the inner product. In this case, the relative representations are equivalent to a Gram matrix, for which it is known that it is possible to reconstruct the original vectors up to isometry. This may be an interesting direction to investigate theoretical claims on absolute reconstruction from generic relative representations.
> >
> > We also plan to include a discussion on this in the camera-ready version of the manuscript, provided that the reviewers agree on the relevance of these considerations.
> >
> >
> > ### References
> >
> > [1] Roeder et al, On Linear Identifiability of Learned Representations, Proceedings of the 38th International Conference on Machine Learning, 2021.
> >
> > [2] Hyvarinen et al, Nonlinear ICA Using Auxiliary Variables and Generalized Contrastive Learning, AISTATS 2019.
> >
> > [3] Khemakhem et al, Variational Autoencoders and Nonlinear ICA: A Unifying Framework, AISTATS 2020.
> >
> > [4] Hyvarinen et al, Unsupervised feature extraction by time-contrastive learning and nonlinear ICA, Neurips 2016.
> >
> > ---
> >
> > We hope that our replies satisfy all the main concerns raised by the reviewer. If it is not the case, we will be happy to further discuss in the remaining time.

---

> > ### Comment · Reviewer_hU9r · 2022-12-04
> > **Thank You and Clarification**
> >
> > Thank you for addressing my questions, I am quite happy with the explanations provided and will raise my score accordingly.
> >
> > To clarify my suggestion regarding the proposed metric (2d in your response):
> > I believe you interpreted it correctly, however it is not directly comparing the absolute embeddings to the relative but rather comparing the *similarities* obtained from the absolute with those obtained from the relative. This seems to be the fundamental aspect you would like to preserve. To reiterate:
> > Given absolute embeddings $A \in \mathbb R^{m_1 \times n_1}$ and $B \in \mathbb R^{m_2 \times n_2}$ and relative embeddings $C \in \mathbb R^{m_1 \times d}$ and $D \in \mathbb R^{m_2 \times d}$, the following characteristics seem desirable and could be checked directly:
> >
> > 1. If $A_i$ and $B_j$ are embeddings of the same vocab item then $C_i \approx D_i$. (Ideally they would be exactly equal, but approximately equal is enough.)
> > 2. The all-pairs similarities $A A^T$ should be similar to those provided by $C C^T$. Obviously we don't need equality here, and perhaps even proportional is too strong. At the very least, however, we'd hope to have similar ordering of the similarities in each row (or column, since it is symmetric). That is to say, the average spearman correlation $\frac 1 {m_1}\sum_k \operatorname{Spearman}(A_k A^T, C_k C^T)$ should be near $1$, and so should $\frac 1 {m_1}\sum_k \operatorname{Spearman}(B_k B^T, D_k D^T)$.
> >
> > If you were able to include evaluations of these two aspects I think that would strengthen the paper even further. That said, I am happy with your responses to my other points, and will adjust my score accordingly. Thank you!

---

> > > ### Author Response · Authors · 2022-12-08
> > > **Absolute-Relative Correlation**
> > >
> > > We thank the reviewer for acknowledging the improvements in our revision and clarifying the proposed metric. We particularly appreciated their detailed feedback and are happy to provide the results of the requested additional experiment.
> > >
> > > The experiment requires checking for two distinct characteristics:
> > >
> > > - The first one,  $\mathbb{C} \approx \mathbb{D}$  if they are embeddings from the same vocabulary, verifies how well the two spaces are aligned by the relative representations. We believe this is already provided in the latest revision of Table 1 in the manuscript. In the table, we report a cosine similarity score of $\approx 0.86$ when comparing the FastTex and Word2Vec relative representations of the same words.
> > > - The second one measures how the self-similarities of each space are preserved by the relative transformation, which could be linked to a notion of metric preservation. This is not something we have already investigated in the work. Thus, in the following, we provide the Spearman correlation coefficient between similarities of the absolute spaces  ($\mathbb{AA}^T$ and $\mathbb{BB}^T$) and similarities of the relative spaces ($\mathbb{CC}^T$ and $\mathbb{DD}^T$).  The experimental setting is the same as the one in Section 4.1; indeed, there is a correlation between the spaces' self-similarities, as [shown in this Figure](https://i.ibb.co/h8sT1PG/spearman.png).
> > >
> > > This new result shows that relative representations not only facilitate latent space communication (characteristic 1) but also preserve the underlying (absolute) latent space metric up to a certain degree (characteristic 2). Thus, we are happy to confirm the reviewer's intuition experimentally. At the moment, we do not have a rigorous theoretical explanation of the reasons behind this property, but we recognize its investigation as an important direction for future works.
> > >
> > > ---
> > >
> > > Let us now consider the [following synthetic example](https://i.ibb.co/rkcq2Nx/anchor-selection.jpg) to understand why measures such as the Spearman correlation between absolute and relative spaces may not always produce perfect scores.
> > >
> > > We start by considering a grid of two-dimensional embeddings, varying only the anchor selection. The first row shows the original absolute embeddings and the chosen anchors, the second row shows the same space after normalization to the unit norm, and the third row shows the relative space computed using the chosen anchors (represented by star symbols). Points in the second and third rows are colored according to the correspondence to the first row.
> > >
> > > There are two interesting observations to be made from this example:
> > >
> > > 1. Even just normalizing the absolute space can cause a loss of information that lowers the Spearman scores.
> > > 2. The anchor selection greatly impacts the correlation between the relative and normalized absolute spaces. In this example, choosing orthogonal anchors (first column) completely preserves the metric and results in a perfect score, but this is not true for the other cases. If the chosen anchors are not representative or do not fully capture the underlying structure of the space, the relative space may not accurately reflect the original space, leading to lower scores.
> > >
> > > Overall, this experiment shows that the normalization and the anchor choice may reduce the correlation between self-similarities.
> > >
> > > ---
> > >
> > > We would like to thank the reviewer once again for their valuable input in improving our work since the absolute-relative relationship is not a straightforward property to analyze or expect. Still, its characterization is arguably desirable, and the reviewer’s comments and questions opened up many new interesting considerations about it. In the next revision, we will incorporate the analysis above in a new paragraph in the appendix, referring to section 4.1.

---

### Official Review · Reviewer_LNFQ · 2022-10-29

**Confidence:** 4
**Clarity, Quality, Novelty And Reproducibility:** See my comments above.
**Correctness:** 4
**Technical Novelty And Significance:** 3
**Empirical Novelty And Significance:** 3
**Recommendation:** 8

**Strength And Weaknesses:**

Strength:

1. The empirical findings in this paper are very impressive and interesting.

2. The paper is extremely well-written and easy to follow.

3. The experiments involve broad domains of machine learning and are of interest to a general audience.

4. The experiment arguments involve rich aspects of analysis and thus are convincing and impressive.


Weakness:

1. The experiments in sec. 4.1 are somewhat weak compared with those in the remained sections; only one case is considered. Adding an extra case of computer vision task may further enhance it.

2. The existence of this phenomenon is well proven, but the source and deep reason for it to happen are not presented as strongly as the former.

3. In Sec.4.1 and Tab. 1, it will be better if the authors can add references and more detailed explanations for the two metrics Jaccard and Mrr. It is a bit hard for me to understand how to calculate them and their roles directly from the current context.

**Summary Of The Paper:**

This paper proposes the opinion that the relative representation, the representation of data described by a fixed set of anchor representations, is invariant among different randomized training factors like seeds, optimization strategies, training steps, and even architectures. To prove their opinion, the authors then prevent extensive studies across a broad range of domains in modern machine learning. The results are impressive and very interesting. The authors show that 1) model trained on datasets sampled from similar domains (like English language) with different distributions may share the same relative representations; 2) the similarity of the relative representations of networks has a positive correlation with their performance; 3) training with relative representations shows comparable performance against the absolute representations; 4) relative representations can be shared by networks of different architectures, training distributions, and seeds, to achieve a valid effect.

**Summary Of The Review:**

Very impressive results in the general area of machine learning, combined with very extensive and strong empirical arguments. Deeper analysis of the sources and reasons of this phenomenon will be appreciated.

---

> ### Author Response · Authors · 2022-11-15
> **Response to Reviewer LNFQ**
>
> We would like to thank the reviewer for the improvement suggestions. Following the advice, we expanded Section 4.1 and addressed the other concerns by:
>
> 1. Adding another **equivalent experiment in the vision domain on the CIFAR10 dataset** in the appendix, with both qualitative and quantitative results. Please refer to Table 8 and Figure 10 in the updated version of the manuscript.
> 2. Incorporating comments and preliminary theoretical analysis on the justification of the phenomenon on which our method builds: please refer to the section “Theoretical analysis” in [answer to *Reviewer hU9r*](https://openreview.net/forum?id=SrC-nwieGJ&noteId=ZDyDqUgvmF).
> 3. **Adding another continuous metric (Cosine Similarity) to compare the latent spaces.**
> 4. Improving the **metric explanations** (Jaccard and MRR). We modified the metrics descriptions in the main paper as follows:
>    > *For each word w, we consider its corresponding encodings x and y in the source and target space. We apply three different metrics to measure their similarity (following a setting similar to Vulic et al. (2020)): (i) Jaccard: the discrete Jaccard similarity between the set of word neighbors x has in source, and the set of word neighbors x has in target; (ii) Mean Reciprocal Rank: measures the (reciprocal) ranking of w among the top-k neighbors of x in the target; (iii) Cosine: measures the cosine similarity between x and y. __Additional details in Appendix A.5.1__*
> 5. Please also refer to the [answer to *Reviewer hU9r*](https://openreview.net/forum?id=SrC-nwieGJ&noteId=HJVw5h5JXSC) for additional **improvements to Section 4.1**.
>
> We hope our replies satisfy all the main concerns raised by the reviewer. If it is not the case, we will be happy to discuss this further.

---

> > ### Author Response · Authors · 2022-12-08
> > **Further analysis and improvements to Section 4.1**
> >
> > We wanted to thank again the reviewer for their positive review and recommendation. We have considered their feedback and addressed their concerns in our previous response.
> >
> > Moreover, we would like to highlight our new [response to Reviewer hU9r](https://openreview.net/forum?id=SrC-nwieGJ&noteId=ShotsAV0Q7R), which expands Section 4.1 and provides a deeper analysis of the relationship between those absolute and relative spaces. We believe this further strengthens the importance of Section 4.1, which we already generalized to the image domain in our previous revision.
> >
> > We would appreciate it if the reviewer could take a moment to review our response and kindly let us know if there are any further comments or questions.
> >
> > Thanks again for the valuable input and for taking the time to review our work.

---

### Author Response · Authors · 2022-11-16
**Revision summary**

We sincerely thank all the reviewers for their constructive feedback on our work, voting unanimously for its acceptance and suggesting actionable modifications to improve the paper further.

Following their advice, we list here the main changes we adopted:
- Generalized experiment in Sec. 4.1 to the vision domain (appendix, Figure 10 and Table 8)
- Added preliminary analysis on anchors selection (appendix A.2, Table 7 and Table 8)
- Improved Jaccard and MRR explanations (main and appendix, Sec. 4.1, A.5.1)
- Added Cosine Similarity metric in Table 1 (main)
- Repeated Table 1 experiments 10 times to report mean and std (main)
- Added Figure similar to Table 1 (left) with other dimensionality reductions (appendix, Figure 9)
- Added clarification on dimensionality reduction in Table 1 (main)
- Added implementation details about the gradient flow through anchors (appendix, A.5.3)
- Emphasized which experiments are trained on relative representations (main, Section 4.3, Section 5.*)
- Improved the related work section (main)
- Fixed fluency and minor typos (main)
- Added theoretical considerations in the section “Theoretical analysis” of the [answer to *Reviewer hU9r*](https://openreview.net/forum?id=SrC-nwieGJ&noteId=ZDyDqUgvmF), which will be added to future revision if deemed relevant by the reviewers.

We provide details and specific replies as answers to each individual review.

UPDATE: 2022-12-08

- Added analysis in the [answer to *Reviewer hU9r*](https://openreview.net/forum?id=SrC-nwieGJ&noteId=ShotsAV0Q7R) on the absolute and relative correlation in Section 4.1 (scheduled for next revision, appendix)

---

### Decision · Program_Chairs · 2023-01-20

**Decision:**

Accept: notable-top-5%

**Justification For Why Not Higher Score:**

N/A

**Justification For Why Not Lower Score:**

Reviewers and AC all agree that the paper is a pleasingly simple idea with high potential impact for enabling practical model re-use, and its findings are convincing and impressive.

**Metareview: Summary, Strengths And Weaknesses:**

The paper proposes the approach of relative representations, which involves representing samples in terms of their latent space distance to a shared set of "anchors", to enable latent space comparison in diverse settings. The authors provide various experiments to show the utility of their approach, particularly that relative representations preserve similarity in a word embedding setting; they do not significantly reduce performance in some image and graph tasks, and a very interesting finding about zero-shot model stitching, where relative representations allow us to transfer latent representations learned on different datasets or architectures, which has promising implications for stitching diverse (e.g. cross-lingual settings), in a simple way.

In general, reviewers greatly appreciated the simple but innovative idea with high potential for impact - its simplicity is especially valuable in enabling such transfer of representations across diverse settings, which can be valuable for model re-use in many practical ways. Reviewers also appreciate the diverse range of experiments which convincingly validate the motivation and effectiveness of the relative representations under various scenarios. All reviewers also found the paper very well written and easy to understand, and its findings to be convincing and impressive. As the authors discuss, the approach of using relative information / anchors has relations to existing work (such as Prototypical Networks / metric learning), but the use-cases, such as zero-shot model stitching, or transferring and comparing across diverse latent spaces, are still innovative.

During the rebuttal stage, the authors made a range of improvements to the paper, which helped to address some of the reviewer concerns and improve the paper, such as adding additional experiments generalizing some findings to the visual domain, other experimental improvements (e.g. repetitions), various improvements to the presentation and clarity, additional experiments on the relationship between absolute and relative representations, etc. In general, the improvements helped to make the experiments more detailed and comprehensive. One reviewer mentioned that adding some theoretical analysis would strengthen the paper; the authors agree that this is certainly an interesting direction for future work (with some promising preliminary directions for analysis mentioned in the rebuttal, such as the relationship to identifiability and nonlinear ICA, and formal characterization of anchor and selection strategy, which can be added as discussions in the paper).

In the end, reviewers and AC all agree that the paper is a pleasingly simple idea with high potential impact for enabling practical model re-use, and its findings are convincing and impressive. Hence, I recommend that it be accepted and highlighted at the conference.

**Note From Pc:**

if the above contains the word "oral" or "spotlight" please see: "oral" presentation means -> notable-top-5% and "spotlight" means -> notable-top-25%. As stated in our emails, we are disassociating presentation type from AC recommendations